# Monitoring of solar spectral ultraviolet irradiance in Aosta, Italy

Ilias Fountoulakis[1], Henri Diémoz[1,2], Anna Maria Siani[3], Gregor Hülsen[4], Julian Gröbner[4]

[1]Aosta Valley Regional Environmental Protection Agency (ARPA), 11020 Saint-Christophe, Italy
[2]Institute of Atmospheric Sciences and Climate, National Research Council, 00185 Rome, Italy
[3]Physics Department, Sapienza Università di Roma, 00185 Rome, Italy
[4]WCC-UV, PMOD/WRC, Dorfstrasse 33, Davos Dorf, CH-7260, SWITZERLAND

*Correspondence to*: Henri Diémoz (h.diemoz@arpa.vda.it)

**Abstract.** A Bentham DTMc300 spectrophotometer is deployed at the station of Aosta–Saint-Christophe, Italy, at the headquarters of the Regional Environmental Protection Agency (ARPA) and has been performing continuous high quality
spectral measurements of the solar ultraviolet (UV) irradiance since 2006. The measuring site is located in the north-western region of the Alps, in a large valley floor at the altitude of 570 m a.s.l., surrounded by mountains. It is very significant to have accurate measurements in such a sensitive environment, since the complex terrain and the strongly variable meteo-climatic conditions typical of the Alps induce large spatial and temporal variability in the surface levels of the solar UV irradiance. The spectroradiometer is also used as a reference of a regional UV network, with additional stations located at
different altitudes (1640 and 3500 m a.s.l.) and environmental conditions (mountain and glacier). In the present study we discuss the procedures and the technical aspects which ensure the high quality of the measurements performed by the reference instrument and the procedures used to characterize the Bentham. The Quality Control/Quality Assurance (QA/QC) procedures are also discussed. We show that the good quality of the spectral measurements is further ensured by a strong traceability chain to the irradiance scale of the Physikalisch-Technische Bundesanstalt (PTB) and a strict calibration
protocol. Recently, the spectral UV dataset of Aosta–Saint-Christophe has been re-evaluated and homogenized. The final spectra constitute one of the most accurate datasets globally. At wavelengths above 310 nm and for solar zenith angles below 75° the expanded (k=2) uncertainty in the final dataset decreases with time, from 7% in 2006 to 4% in the present. The present study not only serves as the reference document for any future use of the data, but also provides useful information for experiments and novel techniques which have been applied for the characterization of the instrument, and the QA/QC of
the spectral UV measurements. Furthermore, the study clearly shows that maintaining a strong traceability chain to a reference scale of spectral irradiance is critical for the good quality of the measurements. The studied spectral dataset is freely accessible at https://doi.org/10.5281/zenodo.4028907 (Fountoulakis et al., 2020).

## 1 Introduction

Although less than 10% of the overall solar electromagnetic radiation that reaches the Earth's surface is in the ultraviolet
(UV), this particular band of the solar spectrum is very significant for life on Earth. Exposure to solar UV radiation is

necessary for living organisms since it triggers a number of beneficial processes (Juzeniene and Moan, 2012;Webb and Engelsen, 2008). However, over-exposure is harmful for the humans and the ecosystems (Caldwell et al., 1998;Häder et al., 2007;Juzeniene et al., 2011;Lucas et al., 2019). Various living organisms, including humans, have been slowly adapted through centuries to the levels of UV radiation at the place where they live. For example, the skin coloration of indigenous

people at different latitudes is a result of the evolutionary process and depends on the levels of the available UV radiation (Jablonski and Chaplin, 2000). However, sun-exposure behaviours of humans are still not optimal in many cases, being responsible for health issues directly related with over- or under-exposure to solar UV radiation. Malignant melanoma (Moan et al., 2008) and cataracts (Taylor et al., 1988;Bourne et al., 2013) are common problems caused by the excessive exposure to solar UV radiation, while hypovitaminosis D (Juzeniene et al., 2011)  is a common problem caused by the

inadequate exposure to UV radiation. Fast changing climate conditions, changes in habits and attitudes of people, and the increase of human migration have meant that many people all over the globe are now exposed to either more or to less solar UV radiation than in the past (Bornman et al., 2019;Cadario et al., 2015;Hintzpeter et al., 2008;Lips and de Jongh, 2018). Thus, many people either need medical supplements or have to drastically change their sun-exposure habits (Lucas et al., 2019). Continuous and accurate monitoring of the levels and the variability of the solar irradiance in the UV spectral region

is necessary for the accurate detection of trends (Glandorf et al., 2005;Weatherhead et al., 1998), as well as estimating different exposure scenarios  in order to inform the public and hence  to  better understand and clarify the balance between the risks and benefits of solar UV radiation under different conditions (Blumthaler, 2018).

Ozone is the main absorber of the (more energetic) photons in the UV-B region (wavelengths 280 – 315 nm), which are more effective on causing both acute (e.g. erythema) and chronic (e.g. DNA damage) problems relative to longer UV

wavelengths. The vast majority of photons with wavelengths shorter than 290 nm, and most photons with wavelengths between 290 and 315 nm are absorbed by ozone in the atmosphere and do not reach the Earth's surface. Photons in the UV-A spectral region (wavelengths 315 – 400 nm) are absorbed less effectively by ozone relative to the photons in the UV-B region. Thus, the overall UV-A irradiance reaching the earth surface is much larger (by more than two orders of magnitude) than the UV-B.  Anthropogenic emissions of Ozone Depleting Substances (ODS) led to enhanced chemical destruction of

stratospheric ozone over high latitudes of the south (Solomon et al., 1986) and the north (Fan and Jacob, 1992;McConnell et al., 1992) hemisphere in spring in the 1980s and the beginning of the 1990s respectively, and subsequently to very high levels of UV-B irradiance at the Earth's surface relative to its past climatological levels (Kerr and McElroy, 1993;Madronich et al., 1998). The concern increased even more after ozone loss occurrences were also observed at middle latitudes which in turn affect solar UV levels at ground (Kerr and McElroy, 1993;Zerefos et al., 1995;Herman et al., 1996;Petkov et al., 2014).

The apparent increase of UV-B irradiance led to the development and deployment of a large number of instruments performing spectral or broad-band measurements of the UV-B irradiance, as well as quantities directly linked to biological effects of the UV radiation such as the ambient erythemal  irradiance (Booth et al., 1994;Gröbner et al., 2006;Schmalwieser et al., 2017). Since the 1980s, improved spectral sensors (compared to those deployed in the 1980s) have been developed, providing more accurate measurements in a wider range of wavelengths extending up to the visible region (e.g. De Mazière

et al. (2018), Zuber et al. (2018)). Furthermore, many international and national inter-comparison campaigns allowed the identification of the main factors introducing uncertainties in the measurements (Bais et al., 2001;Diémoz et al., 2011;Hülsen et al., 2020;McKenzie et al., 1993;Seckmeyer et al., 1994), and subsequently the improved characterization of the instruments, and further reduction of the uncertainties. The good quality of spectral UV measurements performed at different stations around the world is generally ensured by the adoption of standard calibration procedures and comparison with world reference standard instruments which have been developed for this purpose (Gröbner et al., 2006).

Low uncertainty measurements are necessary for the detection of trends in the levels of solar UV irradiance (Bernhard, 2011;Seckmeyer et al., 2009;Weatherhead et al., 1998). According to Glandorf et al. (2005) the magnitude of the detected trends has to be at least larger than the magnitude of the natural variability of the irradiance and the uncertainty in the measurements in order to consider the detected trends reliable. The main sources of uncertainty in spectral UV measurements have been discussed in the studies of Bais (1997) and Bernhard and Seckmeyer (1999). Both studies conclude that uncertainties in spectral UV measurements depend on the characteristics of the measuring instrument and the atmospheric conditions. An inter-comparison between nineteen different instruments in 1997 (Bais et al., 2001) showed that the agreement between well characterized and calibrated instruments is 10% at wavelengths above 300 nm, which is indicative for the expanded (k=2) uncertainty in the measurements. In a more recent study, Cordero et al. (2008) estimated the expanded uncertainty in the measurements of a double monochromator Bentham spectrometer to 9% for wavelengths above 300 nm and SZAs smaller than 30°. Garane et al. (2006) estimated the expanded uncertainty in the measurements of two Brewer spectrophotometers, with a single and a double monochromator, to 13% and 10% respectively for SZAs below 75° and wavelengths above 305 nm. Significant progress has been achieved in the past two decades regarding the methods used for the instrument calibration and characterization, and the quality assurance and quality control (QA/QC) procedures (Fountoulakis et al., 2016;Fountoulakis et al., 2017;Gröbner et al., 2010;Hülsen et al., 2016;Lakkala et al., 2018;Lakkala et al., 2008), which in conjunction to the improvement of the technical characteristics of the instruments (Gröbner, 2003;Pulli et al., 2013) allows spectral measurements with expanded uncertainty of ~2% for wavelengths above 310 nm (Hülsen et al., 2016). In the context of the present study the methods described in the existing bibliography, as well as new methods have been used to characterize a Bentham DTMc300 spectrometer which performs spectral measurements in the UV and visible (VIS) wavelengths of the solar spectrum. The instrument is located at the facilities of Aosta Valley Regional Environmental Protection Agency (ARPA VdA) at Saint-Christophe, Aosta, Italy. The strict QA/QC protocol and the strong traceability chain which ensure the good quality of the spectra are also discussed. Analysis and calculation of the overall uncertainty budget in the spectra, after re-evaluation and homogenization of the measurements confirms that the spectral UV record of Aosta is one of the most accurate datasets globally. Summarizing, the present document deals with the technical aspects of the instrument and the dataset. Analysis and interpretation of the results, including the study of the long-term trends at the Aosta, Saint-Christophe site, will be addressed in a following study.

The paper is separated as follows. In Sect. 2 description of the site location and the technical characteristics of the instruments are provided. The instrument characterization techniques and the methods used for the QA/QC and the

correction of the spectra are described in Sect. 3. The traceability chain is described in Sect. 4. In the same section the different versions of the data are discussed. The main uncertainty sources and the overall uncertainty budget of the spectra are discussed in Sect. 5. Finally, in Sect. 6 the main findings of the study are summarized.

## 2 Location and instruments

### 2.1 Location

The Aosta Valley is an administrative region at the North of Italy. Since it is located in the North-western Alps, its altitude reaches 4800 m a.s.l. (Mt Blanc), with an average of 2000 m a.s.l.. The valley floor where the main settlement (Aosta) is located is surrounded by high mountains, up to 3500 m a.s.l.. The rough terrain, the (spatially and temporally) varying surface albedo and the very complex atmospheric conditions lead to surface levels of UV irradiance which may differ significantly, not only in time but also in space, even within a horizontal distance of a few kilometres (Diémoz and Mayer, 2007). Thus, satellite estimates and model forecasts are uncertain over the region, especially under cloudy conditions (Diémoz et al., 2013;Fountoulakis et al., 2019). Furthermore, high surface albedo and less dense atmosphere lead to extremely high UV indexes (Vanicek et al., 2000) of 11 or more at the highest altitude regions of the Aosta Valley in spring and summer (Casale et al., 2015) and correspondingly high exposure of the tourists and the locals to UV radiation (Siani et al., 2008). Also taking into account that the levels of surface UV radiation are projected to change in the following decades as an adverse effect of climatic change in the sensitive Alpine environment (Bais et al., 2015), it is easy to understand that continuous and accurate monitoring of the solar UV irradiance is essential for the particular region. In order to cover this need, the UV monitoring network of the Aosta Valley which is the first UV monitoring network in Italy, has been created. It includes instruments deployed at three sites at different altitudes  (Fig. 1): Aosta – Saint-Christophe (45.7° N, 7.4° E, 570 m a.s.l.), La Thuile (45.7° N, 7.0° E, 1640 m a.s.l.), and Plateau Rosa (45.9° N, 7.7° E, 3500 m a.s.l.). Although the horizontal distance between them is short (< 35 km), the altitude range is large (3000 m) and they are located in quite different environments (valley bottom, mountain and glacier).

The reference instrument of the network is the double monochromator Bentham DTMc 300 spectrometer with serial number 5541 (hereafter referred as  Bentham5541) located at the Observatory of the Regional Environmental Protection Agency of the Aosta Valley (ARPA VdA), at Aosta-Saint Christophe (hereafter the observatory is referred as AAO).

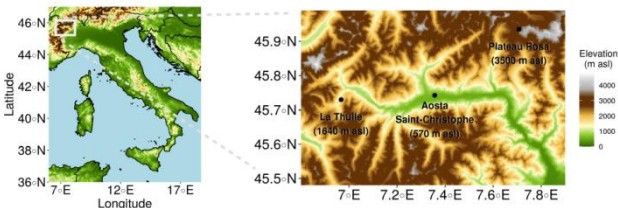

 **Figure 1: The stations of the UV monitoring network of the Aosta Valley**

## 2.2 The Bentham spectroradiometer

In 2004, the Bentham5541 was deployed at the facilities of ARPA VdA and since that year it has been performing spectral measurements in the range 290 – 500 nm (Diémoz et al., 2011). Although the Bentham5541 can also measure photons with wavelengths shorter than 290 nm (i.e. in the range 250 – 290 nm), the solar irradiance with such short wavelengths does not practically reach the Earth's surface. Thus, there is no need to perform measurements below 290 nm. In 2006, the first inter-comparison with the world reference QASUME (Gröbner and Sperfeld, 2005;Hülsen et al., 2016) was performed and allowed the detection of particular operational problems, which were fixed within a few weeks after the inter-comparison. The latter has been developed in the context of the European Commission-funded project Quality Assurance of Spectral Ultraviolet Measurements (QASUME), from which it took its name, aiming at being the reference standard for instruments performing spectral UV measurements around Europe. High quality spectral UV measurements are thus available since 2006, after begin following a strong traceability chain and adopting a strict protocol for the maintenance and the calibration of the system. Currently, Bentham5541 is the reference instrument of the UV monitoring network of the Aosta Valley and its absolute calibration is transferred on a yearly basis to the broadband radiometers which are deployed at the three different stations. In addition to being the reference instrument of the network, the highly accurate UV dataset of the AAO is very useful for climatological studies, and the validation of UV estimates from satellites and radiative transfer models (Fountoulakis et al., 2019;Vitt et al., 2020).

The Bentham DTMc300 is manufactured by Bentham Instruments Ltd (https://www.bentham.co.uk/), with headquarters at Reading, United Kingdom. The spectrometer consists of a pair of symmetrical Czerny Turner monochromators arranged for additive dispersion. The two monochromators are enclosed in a shielded box wherein the temperature is stabilized to 21°C and the relative humidity is kept low with the use of desiccant. The box is also stored in an air-conditioned cabin. Analysis of the internal temperature recordings showed that the temperature in the box wherein the spectrometer and the photon counting system are enclosed is homogeneous within 2°C, and that the temperature of the photon counting system is stabilized within less than 0.5 °C throughout the year (while the ambient temperature usually ranges from about -5°C in winter to 25°C in summer). From 2006 until the summer of 2019 the relative humidity in the box varied between 10% in winter and 60% in summer. Improving the insulation of the box in the summer of 2019 resulted to relative humidity below 40% during the whole year. The variations of the instrument's responsivity with respect to relative humidity were studied and it was found that the latter do not have any detectable effect on the former.

Under usual operational conditions a middle slit with a 1 mm width is used between the two monochromators. The width of the other two (entrance and exit) slits is 0.74 mm. The resulting slit function is approximately triangular with a spectral resolution (FWHM) of 0.54 nm. With the particular setup, the instrument is practically not affected by stray light (Slavin, 1963) as has been also confirmed by comparison with the world reference QASUME (see Sect. 2.4).

The wavelength range and the step of each spectral scan can be determined by the operator. The spectral scans of the solar irradiance are routinely performed in the range 290 – 500 nm with a step of 0.25 nm and a frequency of 1 scan per 15 minutes. The duration of each scan can also change by adjusting the samples per reading (SPR) properly. The wavelength range, the duration, and the frequency of the scans change for example when broad-band radiometers are calibrated at the AAO (wavelength range of 290 – 400 nm, scan time of about 3 minutes, scan repeated without resting time between consecutive spectral scans). During inter-comparison campaigns the measurement settings may also change in order to achieve synchronized measurements.

Solar irradiance enters the instrument through a Teflon diffuser which is covered by a UV-transmittable quartz dome and is enclosed in a thermally stabilized case. The whole system (case, diffuser, dome, desiccant) is hereafter referred as the optical head (OH) of the instrument. A heater has been installed in the OH, which has been regulated in order to keep the temperature above 32°C. The OH is the UV-J1002-REG system which was purchased from Schreder CMS (Lofererstrasse 32, 6322 Kirchbichl, Austria). The light entering the diffuser is guided into the spectrometer using an optical fiber. As already discussed, the box containing the monochromators and the PMT is kept in a thermally stabilized cabin. The photomultiplier tube (PMT) is a DH-10 side window Bialkali deployed in the photocurrent mode, with a maximum sensitivity in the UV. An amplifier (267 programmable D.C. amplifier) amplifies the signal by a factor ranging between 1 and 10 and the signal is finally recorded by a computer (PC). More specifications regarding the standard characteristics of the spectrometer can be found in the web-page of the manufacturer (https://www.bentham.co.uk/products/components/dtmc300-double-monochromator-39/).

**2.3 Other UV monitoring instruments of the network**

Since 2007 spectral UV measurements in the range 290 – 325 nm have been performed by the single monochromator MKIV type Brewer with serial number 066 (Brewer#066), which measures the solar irradiance with a step and a resolution of ~0.5 nm. Brewer#066 was moved to AAO from the Environment Institute of the European Union-Joint Research Centre, Ispra (45.8° N, 8.6° E, 240 m a.s.l.) in 2007. At the AAO, the UV-A and erythemal (CIE, 1999;Webb et al., 2011) irradiances have been measured by a UVS-AE-T dual-band Kipp & Zonen radiometer since 2005. A second radiometer of the latter type was installed at the high altitude station of Plateau Rosa in 2007 and measures the same radiometric quantities. One more radiometer (Yankee UVB-1) at the station of La Thuile provides measurements of the erythemal irradiance since 2005. In 2019 the UVS-AE-T at AAO stopped working and has been replaced by a UV-E Kipp & Zonen radiometer. Since 2006, the Bentham5541 is the reference for all broadband UV monitoring instruments of the network with the exception of the Brewer#066. The Brewer#066 is calibrated independently every two years from International Ozone Services Inc. (IOS) (https://www.io3.ca/) using a 1000W lamp which in turn is traceable to the National Institute of Standards and Technology (NIST) (https://www.nist.gov/). Measurements of the total column and the profiles of different atmospheric components are also available at the station (Diémoz et al., 2019a;Diémoz et al., 2014;Diémoz et al., 2019b;Siani et al., 2018), and can be used in order to better understand how UV radiation interacts with the atmospheric components in the complex environment

of the Aosta Valley. Radiative transfer models are also employed to estimate the solar UV irradiance in the whole domain of study, and to predict the UV Index for the following days. This informative parameter is provided by a bulletin for the general public of the health risk of UV radiation exposure (http://www.uv-index.vda.it).

## 2.4 The world reference QASUME

The Bentham5541 is traceable to the scale of spectral irradiance established by Physikalisch-Technische Bundesanstalt
(PTB). Regular on-site audits with the world reference QASUME further ensure the traceability to PTB, as well as the accuracy and the good quality of the measurements (see Sect. 4.1). QASUME is a transportable reference spectroradiometer for measuring spectral solar ultraviolet irradiance. The World Radiation Center at the Physical Meteorological Observatory in Davos (PMOD-WRC) is responsible for the maintenance and the continuous improvement of the system. Since 2002 it is the reference standard for many stations around Europe. The spectrometer of QASUME is a commercially available
Bentham DM-150 double monochromator. Initially the entrance optic was a CMS-Schreder, Model UV-J1002 (same as the one deployed in the Bentham5541), which since 2016 has been upgraded, and the Bentham D6 input optic is used. Optimization of characterization methodologies and implementation of the new optic have reduced the expended uncertainties of solar spectral UV irradiance measurements from 4.8% in 2005 to 2.0% in 2016 in the spectral region above 310 nm (Hülsen et al., 2016). More information about QASUME can be found in the referred bibliography (Gröbner and
Blumthaler, 2007;Gröbner et al., 2006;Gröbner et al., 2005;Gröbner and Sperfeld, 2005;Hülsen et al., 2016).

## 3 Characterization of the Bentham5541 and correction of the measurements

Since 2006, there have been efforts to determine the individual instrumental characteristics of the Bentham5541 and apply proper correction factors to the spectra in order to take into account the instrumental characteristics which can introduce biases or errors in the dataset. In many cases it is not possible to correct the measurements, and the remaining biases or errors
are taken into account in the calculation of the uncertainty budget of the dataset (see Sect. 5). The instrumental characteristics mostly affecting the measurements, as well as the procedure followed for the determination of their effects and the subsequent correction of the measurements are described below.

### 3.1 Dark signal and amplification

The dark signal consists of electrical charges generated in the detector when no photons enter the system. The dark signal
($D$) is measured before the beginning of each spectral scan when the shutter is closed and no photons enter the PMT. It is generally in the order of ±0.1 pA. Although the dark signal depends on temperature, it does not practically change during the spectral scan under usual operational conditions. The recorded signal from the sources used for calibration is larger by three orders of magnitude or more, relative to the dark signal for wavelengths above 280 nm. Thus, the uncertainties in the determination of the calibration factor related to the dark signal can be considered negligible. The role of dark signal may be

more significant for very short wavelengths (shorter than 305 nm when SZA is larger than 75° or for smaller SZAs when the sky is overcast). At 305 nm the irradiance is higher than the dark signal by at least two orders of magnitude, even at SZAs around 85°. Thus, for wavelengths longer than 305 nm and for SZAs below 85° we can consider that the uncertainties related to the dark signal correction are negligible relative to the overall uncertainty budget of the measurements. Of course, as the signal decreases approaching 0 the uncertainty related to the dark signal becomes more important. The uncertainties related to the amplification ($A$) of the recorded signal ($I_0$) are also negligible. The corrected signal ($I_1$) at each wavelength is calculated from Eq. 1.

$$I_1 = (I_0/A) - D \ (Eq. 1)$$

## 3.2 Temperature of the Teflon diffuser

Ylianttila and Schreder (2005) studied the change of the transmissivity of Teflon diffusers (including diffusers of similar thickness as the Teflon diffuser used by Bentham5541) with respect to temperature. As discussed in the study of Ylianttila and Schreder (2005) the transmissivity of the Teflon diffuser to solar radiation changes with respect to temperature. The temperature of the Teflon diffuser is recorded using a thermistor which has been installed in the OH and is in contact with the bottom surface of the diffuser. The temperature stabilization system can warm up, but not cool down the diffuser. Thus, in summer the diffuser is getting warmed above the temperature stabilization point (32°C), and around local noon its temperature may reach 40°C. In winter, the temperature stabilization system cannot always fully compensate for the very low ambient temperatures. Thus, the temperature of the diffuser may be very low (~15°C) in the morning. Around noon (in winter) the temperature increases and is usually closer to the desired level. The transmissivity of Teflon diffusers similar to that of Bentham5541, increases fast by 3% from 15°C to 22°C, and thereafter decreases gradually by 2% between 22°C and 45°C (Ylianttila and Schreder, 2005). The dependence on wavelength is very small and thus it is considered negligible and is not taken into account for the correction of the measurements. In particular, the signal $I_1$ calculated from Eq. 1 is corrected for the effect of temperature using Eq. 2:

$$I_2 = I_1/c \ (Eq. 2)$$

Where, $I_2$ is the corrected signal, and $c$ is the correction factor. The correction factor is calculated from Eq. 3:

$$c = a \cdot \theta + b \ (Eq. 3)$$

Where $a$ and $b$ depend from temperature of the diffuser $\theta$. More specifically:
For $\theta$<15°C, $a = -0.00075, b = 0.99675$
For $\theta$>=15°C and $\theta$<22°C, $a = 0.0043, b = 0.9104$

For $\theta$>=22°C, $a = -0.00083, b = 0.9867$

The coefficients $a$ and $b$ were taken from Ylianttila and Schreder (2005).

Before 2017, no correction was applied to the spectra although temperature was recorded at the beginning of each scan (a post-correction has been applied on the new dataset as discussed in Sect. 4.2). Since the 5[th] of May 2017 the temperature is recorded several times during the spectral scan and all spectra are automatically corrected for the effect of temperature using Eq. 2. As discussed more analytically in Sect. 4.2 the difference between the corrected and the uncorrected spectra for the effect of temperature is up to 3% in winter, and up to 1% in summer. In order to validate the accuracy of the used correction

factors two different experiments were performed.

First experiment: In a summer day the irradiance from two 200 Watt lamps was measured in the morning, at noon, and in the early and late afternoon. The temperature of the diffuser varied between 34°C and 40°C in the day. After correcting the lamp spectra for the effect of temperature the agreement between them (after applying a 10 nm moving average smoothing filter to compensate for noise) was better than 0.5% for all wavelengths.

Second experiment: On a different day we transferred the OH inside the optical laboratory and switched off the temperature stabilization system of the Bentham's OH. We cooled the air in the room down to 18°C during the night and in the morning we begun performing measurements of the irradiance from different (1000 Watt and 200 Watt) lamps. The temperature stabilization system remained switched off and the air in the room was gradually warmed up while measurements were performed at different temperatures. A thermal camera was used to monitor the temperature of the diffuser complementary to

the thermistor recordings. Before taking each picture with the camera, the dome that covers the diffuser was removed. When temperature reached 28°C the stabilization system was turned on and used to increase the temperature of the diffuser up to 32°C. Then, an electrical heater was used in order to further increase the temperature in the room, and consequently the temperature of the diffuser, up to 41°C. The accuracy of the correction was verified by the fact that the agreement among the corrected lamp spectra (after applying the same smoothing filter as before) recorded at temperatures between 18°C and 41°C

was better than ±0.5% for wavelengths above 350 nm, and ±1% for shorter wavelengths, without any obvious dependence of the remaining differences from temperature.

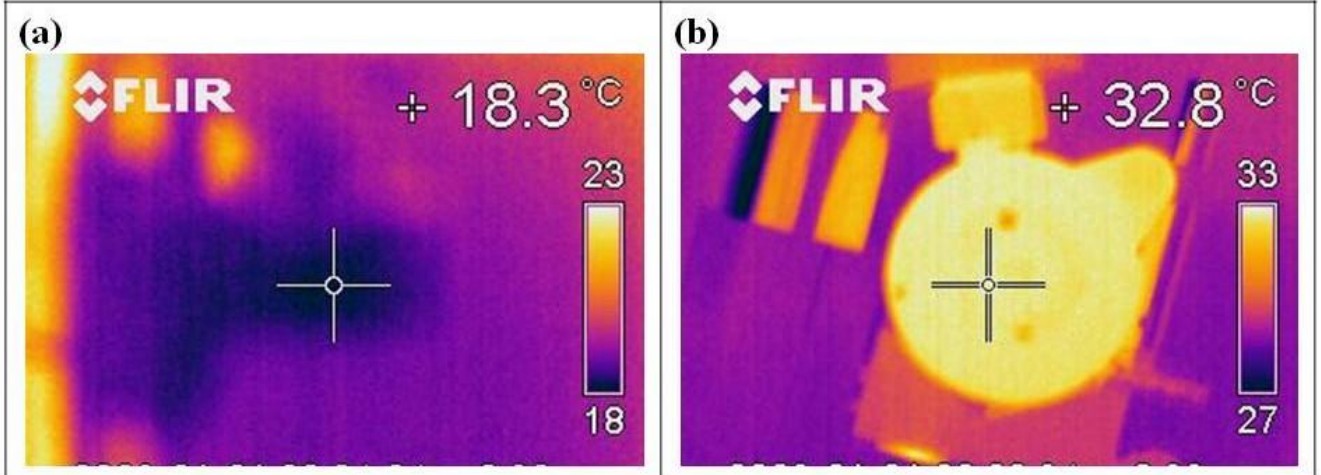

Figure 2: Temperature of the OH (a) in the morning with the internal heating turned off, and (b) with the internal heating turned on.The temperature stabilization system warms the diffuser uniformly as shown in Fig.2, where two different pictures taken with the thermal camera (with the temperature stabilization system off (a) and on (b)) are presented. In both the homogeneity in the temperature of the OH is obvious. Fountoulakis et al. (2017) showed that during measurements of the irradiance using 50 Watt lamps (distance between the lamp filament and the diffuser was 5 cm) with a Brewer spectrophotometer, the temperature of the Teflon diffuser was inhomogeneous, and was higher by 5-6°C at the center relative to the edges of the diffuser. At AAO the irradiance from 200 Watt lamps is measured using a KS-J1011 portable field calibrator (PFC) provided by Schreder CMS (http://www.schreder-cms.com/en/). The distance between the lamps' filament and the diffuser is 12 cm for the original setup, and can be increased to 30 cm with the use of an extender. Measurements with the thermal camera in AAO showed that even when the extender is not used during measurements of the irradiance from the lamps, the temperature variations over the diffuser are below 2°C. In all cases discussed above the temperature recorded by the thermistor, and the temperature measured by the thermal camera agreed within 1°C.

When the short setup (PFC without the extender) of the calibrator is used, the diffuser is warmed up during measurements of the lamps' irradiance. In summer days temperatures up to 46°C have been recorded. Thus, since mid-2018 the extender has been used regularly in order to increase the distance between the lamp and the diffuser to ~ 30 cm. Although using the long setup results to increased noise because of the weaker signal (thus, slightly higher uncertainty in the retrieved calibration factors) the diffuser is practically not warmed during measurements.

## 3.3 Angular response

### 3.3.1 Characterization

Despite the progress that has been achieved regarding the improvement of the angular response of the Teflon diffusers commonly used as entrance optics, there are still imperfections that induce uncertainties in the measurements, even when

high quality entrance optics are used (as in the case of Bentham5541). The angular response of the diffuser was measured at
300 the facilities of the PMOD –WRC in 2014 and the cosine error of the diffuser defined by the ratio between the measured angular response and the ideal response is presented in Fig. 3. The characterization methodology was similar to that described in the studies of Antón et al. (2008) and Bais et al. (1998).

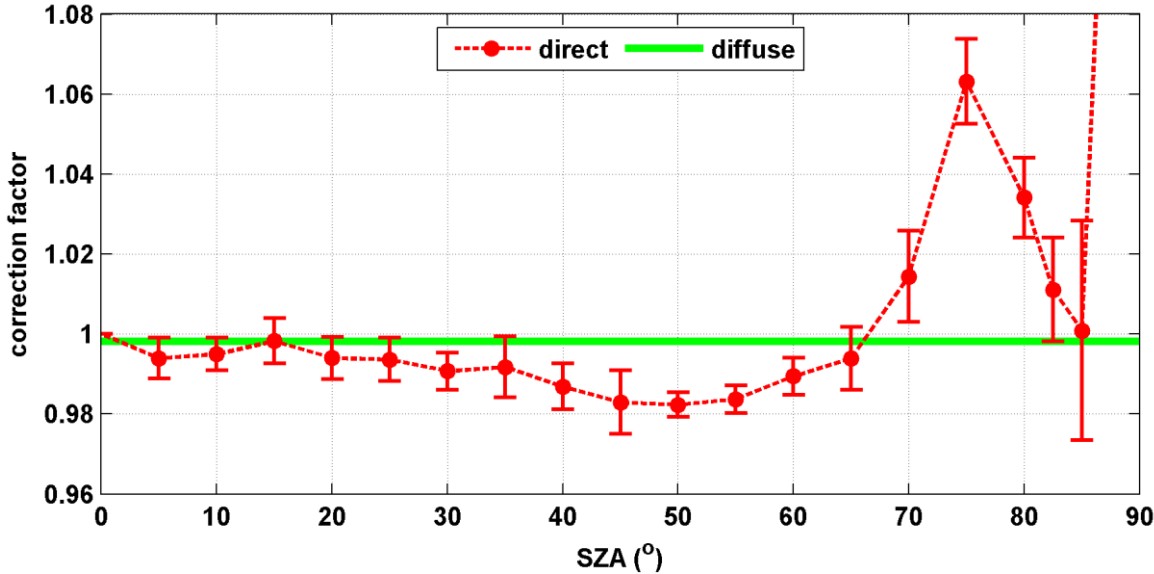

**Figure 3: Cosine error of the diffuser defined by the ratio between the measured angular response and the ideal response, for**
**illumination by a point source (red) and for isotropic illumination (green). Red dots represent the average cosine error (for all four azimuth directions) for illumination by a point source. Error bars represent the corresponding standard deviation. The ratio (for illumination by a point source) at 88° is 1.38 and is outside the limits of the y-axis.**

The characterization was performed at 320 nm for SZAs 0 - 90° with a step of 5° at four different azimuth directions. These directions, relative to the orientation of the diffuser during usual operational conditions, are North, South, West, and East.
The angular response of Teflon diffusers similar as the one used for Bentham5541 is generally getting worse (thus the error becomes larger) with increasing wavelength. However, Bernhard and Petkov (2019) studied this effect for a similar Teflon diffuser and found that differences become important for wavelengths longer than 500 nm while in the range 300 – 500 nm the differences are within the noise of the measurements.

Assuming that the diffuse irradiance is isotropic (see e.g. Gröbner et al. (1996)), the diffuse component of the solar
irradiance is underestimated by 0.2% according to the results presented in Fig. 3. For the direct component the % error is less than 2% for SZAs below 70°, increases up to 6% at 75°,  then decreases again up to 85°, and increases fast thereafter, up to 38% at 88°. However, even in the visible range of the solar spectrum, the direct component has a very small contribution to the overall solar irradiance at SZAs above 85° making the effect of the cosine error at such large SZAs insignificant.

For the same SZAs, the % cosine errors at the four different directions differ to each other by less than 2% (±1% relative to
320 the average) up to the SZA of 80°. The small differences which are within the uncertainty of the characterization show that

the cosine response of the diffuser is practically independent from the azimuth direction. For SZAs between 80° and 88° the differences between the errors at different planes gradually increase reaching a maximum of about ±5% relative to the average, at 88°. However, the characterization uncertainties above 80° are large due to the low signal and the finite width of the light beam used for the characterization, as well as the effect of errors in the direction of the incident light beam (at 88°, an error of 0.1° leads to a corresponding difference of 5% in the measured irradiance) which introduce additional uncertainties. Thus, (at least part of) the difference is again more likely due to the characterization uncertainties than real differences in the cosine response.

### 3.3.2 Modelling the errors due to angular response

The error in the measurements of global irradiance depends on the ratio between the direct and diffuse components (Bais et al., 1998). Thus, in order to estimate the error due to the non-ideal angular response of the instrument, simulations of the direct and the diffuse components were performed for SZAs between 0° and 90° using the model UVSPEC included in libRadtran package (Emde et al., 2016). Simulations were performed for typical columnar ozone of 320 DU, and aerosol optical depth (AOD) equal to 0.05 and 0.2 (at 500 nm). AOD was scaled to shorter wavelengths using an Ångström exponent equal to 1. A six-stream approximation was used for pseudospherical atmosphere and standard atmospheric profile (Anderson and Division, 1986). We investigated the results for two different altitudes: 570 m, and 1590 m. The former is the altitude of the AAO, while the latter is the altitude of the PMOD –WRC (46.8°N, 9.8°E, 1590 m a.s.l.). At high altitude sites such as Davos the contribution of the direct component to the global solar irradiance under clear skies is stronger relative to lower altitude sites (such as Aosta) due to weaker Rayleigh scattering and the usually negligible attenuation by aerosols. The larger direct component of the solar irradiance is responsible for larger uncertainties at SZAs between 70° and 80° where the cosine error for the direct beam is more pronounced (Sect. 3.3.1). Thus, simulations were also performed for Davos in order to have an estimate of the effect of errors related to the angular response at such high altitudes. These results are also useful for the discussion carried out in the following sections since in July of 2014 the Bentham5541 took part in an inter-comparison campaign at Davos (see Sect. 4.2.6).

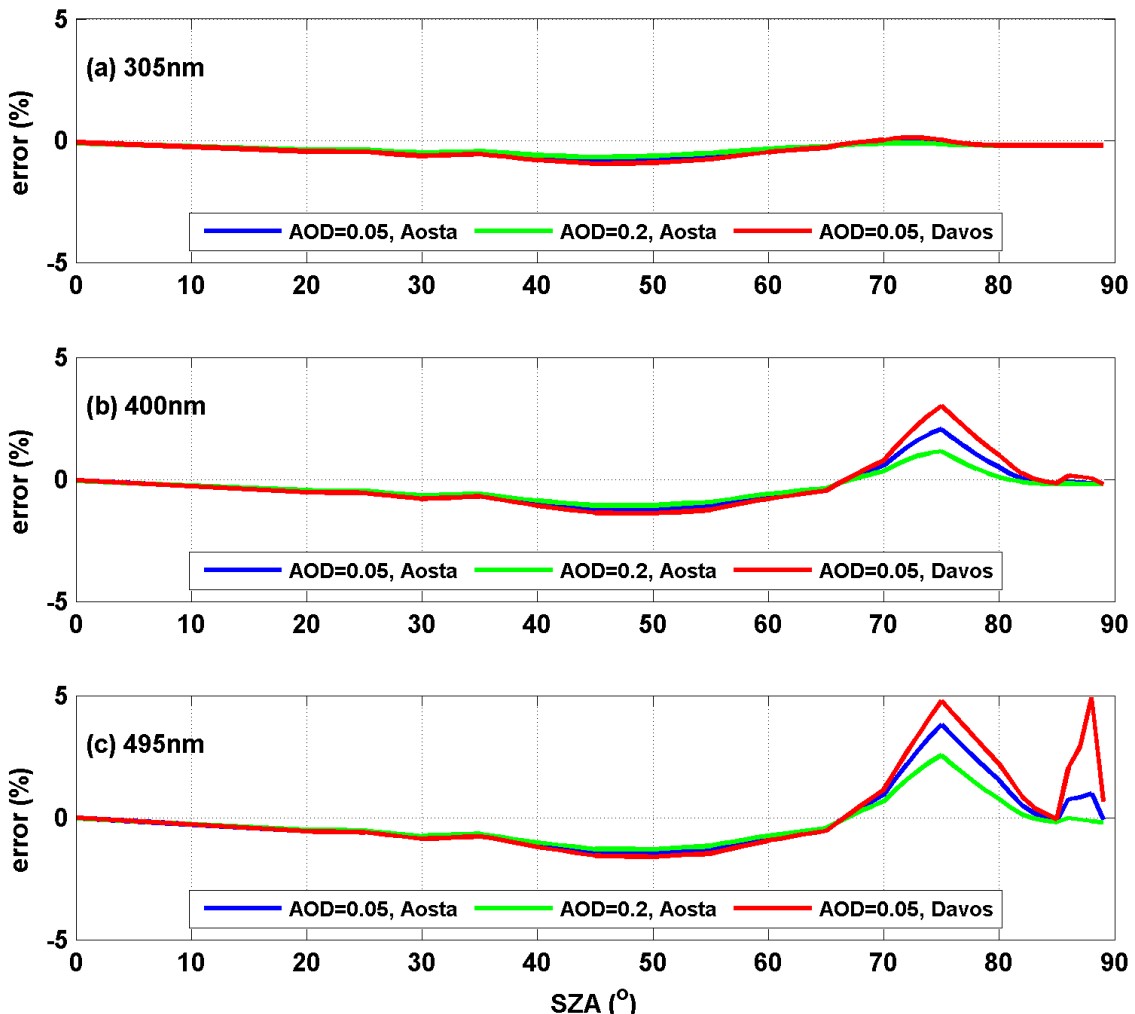

**Figure 4: Modeled error in the measured global irradiance for three different wavelengths: (a) 305 nm, (b) 400 nm, and (c) 495 nm. The results are for AAO (for AOD at 500 nm equal to 0.05 and 0.2) and the PMOD-WRC (for AOD at 500 nm equal to 0.05).**

The results for 305, 400, and 495 nm and AOD at 500 nm equal to 0.05 and 0.2 (for Aosta), and 0.05 (for Davos) are presented in Fig. 4. The results for AOD=0.2 are not presented for Davos since the particular value is unrealistic for that station. For wavelengths below 350 nm the error remains always below 1% because at such short wavelengths the diffuse component of the irradiance is dominant for SZA larger than 70° where the angular response is worse. For longer wavelengths the errors become more important. For the SZA of 75° and 495 nm the error is 4% for Aosta and 5% for Davos when AOD is 0.05. Taking into account the uncertainty in the characterization of the diffuser, and the fact that the cosine response at 495 nm may be slightly worse than that measured at 320 nm we estimate that the maximum error may be slightly

larger, i.e. ~5% for Aosta and ~6% for Davos. For SZAs larger than 85° the contribution of the direct irradiance at Aosta is

very small even for wavelengths near 500 nm, resulting to errors below 2%. However, at Davos the contribution of direct irradiance is still significant resulting to errors in the order of 5%.

Systematic, relatively accurate correction for the effect of non-ideal cosine response can be achieved only if the ratio between the direct and the diffuse solar irradiance that reach the diffuser is known. However, measurements of the direct component of spectral irradiance are not available. The use of a modelling approach as suggested by other studies (Lakkala

et al., 2018) in order to calculate the ratio, would be also highly uncertain because of the varying cloud, albedo and aerosol conditions, which cannot be easily modelled. In order to reduce the uncertainty related to the imperfect cosine response of the diffuser, a new Teflon diffuser with improved angular characteristics is planned to replace the one which is currently used.

### 3.4 Alignment of the optical fiber and levelling of the fore-optics

The light that enters the diffuser is transmitted to the spectrometer through the optical fiber. There are two metallic terminations at the two ends of the fiber. The light passing through the Teflon diffuser enters the fiber through a small aperture at one end and exits the fiber through a second aperture on the other end. The aperture at the exit has to be aligned with the entrance of the monochromator so that the maximum possible light enters the two monochromators. However, non-optimal alignment of the termination at the exit of the fiber is not a big problem since it is implicitly taken into account

during the instrument absolute calibration.

Optimal alignment of the metallic termination relative to the reference plane of the diffuser at the entrance optics is more important. Improper positioning may result to inhomogeneous response of the diffuser with respect to the zenith and/or the azimuth angle of the sun. The errors are again more significant for longer wavelengths for which the direct component is stronger, i.e. for measurements in the UV-A and VIS regions. During inter-comparisons with QASUME, continuous

measurements are performed while the OH of each instrument is rotated by 180° in order to detect possible azimuthal dependence of the response due to the imperfect alignment of the termination of the fiber. Based on the results of the inter-comparisons with QASUME reference in the period 2006 - 2019, we estimate that the errors related to the misalignment of the fiber optics are usually below ±0.5%, but are higher in particular periods of a few months (up to 3.5% for VIS and 2% for UV) as discussed in Section 5.

The levelling of the fore-optics (i.e. the Teflon diffuser) is checked on a weekly basis. The levelling is accurate within ± 0.1° and the corresponding error is generally negligible with the exception of a particular period. From 2006 until 15 January of 2015 the diffuser was levelled using the bubble level which is adjusted to the OH. Thereafter a levelling jig is used. A particular problem led to a larger mis-leveling of the diffuser and correspondingly to larger errors in the measurements in the period 07/07/2014 – 14/01/2015. In particular, the OH was disassembled and re-assembled before an inter-comparison

campaign in Davos in 2014 (see Sect. 4.2.6). After re-assembling the OH the position of the bubble level on the OH changed. Thereafter, using the bubble of the OH to align the diffuser leads to a mis-levelling of +0.7±0.1° at an azimuth

direction of ~340° (~20° from the south, towards the west). After using the levelling jig the particular problem has been solved. Due to this problem the variability in the ratio of synchronous clear-sky measurements between QASUME and AAO was wavelength dependent, varying from ± 1% at shorter wavelengths to ± 4% near 500 nm during the 2014 inter-comparison campaign at Davos, which is unusually large.

### 3.5 Change in responsivity after exposure to high radiation levels

Before 24 July of 2006 the AAO measurements were performed with the high voltage (HV) of the PMT set to 670 V. With the particular HV setting the PMT was exposed to signal levels which were well beyond its optimal operational range resulting to large diurnal variability (changes in the order of 10% in summer days) in the responsivity of the PMT. Thus, the HV was thereafter lowered to 400 V. Although lowering the HV so much led to increased noise in the measurements, it also solved the issue of changing responsivity as has been confirmed after performing the experiments described below.

### 3.5.1 Change in responsivity during usual operating conditions

The maximum photocurrent measured by Bentham5541 (in the noon of cloud free summer days) is in the order of ~300 nA. During the Davos inter-comparison campaign in 2014 the measured signal reached higher values of ~500 nA. It is noteworthy that before lowering the high voltage of the PMT (in July, 2006) the photocurrent was in the order of 15000 – 20000 nA. Two problems which have affected the Bentham5541 spectroradiometer during its regular operation before July 2006 are the following:

1. The responsivity gradually changed after a few days of consecutive measurements with the PMT being exposed to high signal levels.
2. The responsivity was changing during the day following the intensity of the recorded signal. In this case the responsivity of the PMT decreased after exposure to very high signal. Then it gradually increased again until the next spectral scan begun. The responsivity in this case changes during the day depending on the resting time between consecutive scans, and the maximum intensity of the recorded signal during each scan.

In order to investigate the first issue, consecutive spectral scans, without leaving any rest time between them, were performed in the range 280 – 400 nm between 27 June 2018 and 6 July 2018. The sky during most of these days was cloudless and the levels of the UV irradiance were very high (noon UV index between 7 and 9). The irradiance of two 200 Watt lamps was measured at the beginning and at the end of the 10-day period, and for both lamps the difference was less than 0.5%. The diffuser temperature during the measurements performed in the two different days was similar (33.5 - 35°C) ensuring that the effect of differences in temperature has not played any significant role.

In order to investigate the second issue, measurements with 200 Watt lamps were performed in the early morning, noon, and late evening of different summer days. In all cases the lamps' irradiances were measured directly after solar scans. The measurements were repeated using different schedules for the spectral scans (different spectral range and different resting

times between the spectral scans). In all cases, after correcting the measurements for the effect of temperature (of the Teflon diffuser) the differences were again less than 0.5%.

### 3.5.2 Change in responsivity after exposure to unusually high radiation levels

As a second step we tried to investigate if the responsivity of the instrument changes for signal levels higher than usual. In order to achieve higher signal levels we performed measurements using wider (or totally removing) slits. Measurements were performed in the period 18 – 20 September 2018. Sky was cloudless during most of the day for all three days. We used two different setups:

**Setup 1:** Standard entrance slit (0.74 mm), standard intermediate slit (1.00 mm), no exit slit
**Setup 2:** All slits were removed (no entrance, intermediate or exit slit)

When the first setup was used the maximum photocurrent was ~1100 nA (~ double than the maximum photocurrent in Davos, ~3-4 times higher than maximum photocurrent in Aosta). In the morning no measurements were performed until ~8:00 LT. Then, the irradiance of two different 200 Watt lamps was measured. After performing the two scans, consecutive measurements of the solar irradiance in the range 290 – 400 nm were performed for ~3 hours (again without letting the PMT to rest between consecutive scans) using the same schedule as during QASUME inter-comparison campaigns. Then, spectral scans of the irradiance of the same two lamps were repeated (at ~13:30 LT). After correction for the effect of temperature the difference between the morning and noon lamp scans was again less than 0.5%. However, in this case the responsivity of the Bentham5541 increased while we were measuring the lamps' irradiance at noon. The noon lamp scans lasted 40 mins (including the resting time between them), during which the responsivity increased by ~0.5%. Since the increase is small and well below the instrumental uncertainties, we decided to perform one more experiment in order to further investigate if it is real.

Using the same (first) setup we continued performing spectral scans of the solar irradiance for ~ 1 more hour after finishing the lamp scans. After 1 hour of measurements we interrupted a spectral scan (at 370 nm when very strong signal had already been recorded) and within a few seconds we begun performing measurements of the irradiance from a 200 Watt lamp at 330 nm (photocurrent ~5nA). The lamp signal at the wavelength of 330 nm is weak, thus we assume that it does not affect the responsivity of the PMT. Indeed, the responsivity increased by ~0.5% in one hour after interrupting the spectral scans and remained relatively stable thereafter. In the first 40 minutes, the increase was 0.3 – 0.4% which confirms the findings of the first experiment.

In order to investigate if even higher signal has a stronger impact on the performance of the PMT we removed all slits and repeated the two experiments. The maximum photocurrent was ~2500 nA (~ 5 times higher than the maximum photocurrent in Davos, 9 – 10 times higher than maximum photocurrent in Aosta). In this case, the difference between the morning and the first noon scan of the irradiance from a 200 Watt lamp was ~3% (the responsivity decreased by ~3% from morning until noon). Repeating the second experiment (interrupt spectral scans of the solar irradiance and perform continuous lamp measurements at a particular wavelength) resulted to increasing responsivity (by ~2% in 100 minutes). In the 100 minutes of

the measurements there was no sign of stabilization, showing that the responsivity would probably keep increasing if the measurements had continued for more than 100 minutes.

According to these findings, the responsivity of the PMT  is not affected from the level of the recorded signal for usual operational conditions with the HV set to 400 V. All measurements performed before the 24[th] of July 2006 are however less accurate since the very high photocurrent was affecting the responsivity of the PMT. Thus, they have not been included in the Level 1.5 and Level 2 datasets.

### 3.5.3 Linearity

Comparison of the spectral measurements from Bentham5541 with simultaneous measurements of QASUME during recent inter-comparison campaigns (2015, 2017, 2019), as well as with measurements from broad-band instruments operating at AAO did not yield any sign of detectable non-linearity of the Bentham5541. Thus, even if there is any non-linearity effect the relative uncertainty is very small relative to the overall uncertainties in the measurements.

### 3.6 Wavelength shift

In order to correct the near real time spectra for the effect of wavelength shift, the SHICrivm algorithm (Slaper et al., 1995) has been used, which results in a wavelength accuracy in the range 305 – 500 nm in the order of ±0.02 nm after correction. The uncertainties in the processed spectra have been investigated and discussed in past studies (Diémoz et al., 2011;Gröbner and Sperfeld, 2005) and are presented in Sect. 5. Thus, no relative investigation has been performed in the context of the present study. The re-evaluated spectra have been processed using the MATshic (Egli et al., 2016) instead of the SHICrivm algorithm. However, the results from the two algorithms are very similar when the same settings are used. Comparison between spectra which have been processed using both algorithms also confirmed their compatibility.

### 3.7 Other issues

Between 2018 and 2020, there were five different periods of a few days (from 5 to 15) during which the Bentham5541 (for different reasons) was not performing measurements of solar irradiance. Performing measurements with 200 and 1000 Watt lamps during these periods showed that the responsivity of the Bentham5541 gradually decreases (by 2-4% in a few days) and then remains low as long as the instrument is inactive. Although it was not possible to detect the exact reasons that are responsible for this change in the responsivity, we noticed that the responsivity returns to the pre-resting period levels directly after setting back the instrument to regular operation and performing a few spectral scans. Thus, this issue was also not investigated in more depth. Although the variability in relative humidity and temperature inside the shielded box which contains the spectrometer may affect the electronic and mechanical parts of the instrument, it was not possible to detect any clear correlation between the variability of the particular parameters and the responsivity. In any case, performing the calibrations on a monthly basis ensures that the responsivity does not change by more than ~1.5% between consecutive calibrations.

## 4 Calibration, traceability and data versioning

### 4.1 Traceability chain

Since the beginning of high quality spectral measurements in 2006, a strict protocol for the maintenance and the calibration of the system has been adapted and a strong traceability chain has been followed. On a monthly basis, the Bentham5541 is calibrated by measuring the irradiance from two 200 Watt lamps (provided by Schreder CMS and adjusted for the KS-J1011 PFC, hereafter referred as KS lamps). The two lamps are chosen from a set of three lamps (lamps are rotated in order to accurately detect possible problems). No time interpolation is applied for the calculation of the calibration factors between consecutive calibrations. Possible drifts or unexpected changes in the responsivity between consecutive calibrations are taken into account in the overall uncertainty budget of the measurements. In the period 2006 – 2016 two of the 200 Watt lamps were brought on a yearly basis to the European UV Calibration Center (EUVC) hosted by PMOD – WRC, where their spectral irradiances were measured. The EUVC irradiance reference is realised through the average of seven secondary standard lamps calibrated by PTB between 2002 and 2009 against the primary reference for spectral irradiance, blackbody BB3200pg (Gröbner and Sperfeld, 2005;Hülsen et al., 2016).

The type of the fore-optics of Bentham5541 and QASUME was the same in the period 2006 – 2016 allowing the characterization of the full calibration setup used in Aosta (measurements in Davos were performed using the PFC of AAO). After the return of the lamps from Davos, they were used as reference in order to re-calibrate the third 200 Watt lamp (which had not travelled to Davos) at the AAO.

After the fore-optic of QASUME was upgraded in 2016 a direct calibration transfer of the 200 Watt lamps using the PFC at Davos is impossible. Thus, a new optical laboratory has been set up at the AAO in 2018 and two new 1000 Watt FEL lamps have been purchased, and are currently used as reference in order to re-calibrate the three 200 Watt working standards. The two 1000 Watt FEL lamps are again brought to PMOD – WRC on a yearly basis in order to be recalibrated. Then, the spectral irradiance of each of the 200 Watt lamps is measured in the optical laboratory against the 1000 Watt lamps. Regular inter-comparisons with the QASUME reference every one or two years complement and close the traceability chain. The Bentham5541 is finally used as a reference instrument for the broad-band radiometers of the UV monitoring network of the Aosta Valley. Side by side measurements are performed by Bentham and radiometers for 1-3 weeks (depending on the weather conditions) in order to transfer the calibration from the former to the latter.

The two different traceability chains for the periods 2006 – 2018 and 2018 – present are presented graphically in Fig. 5.

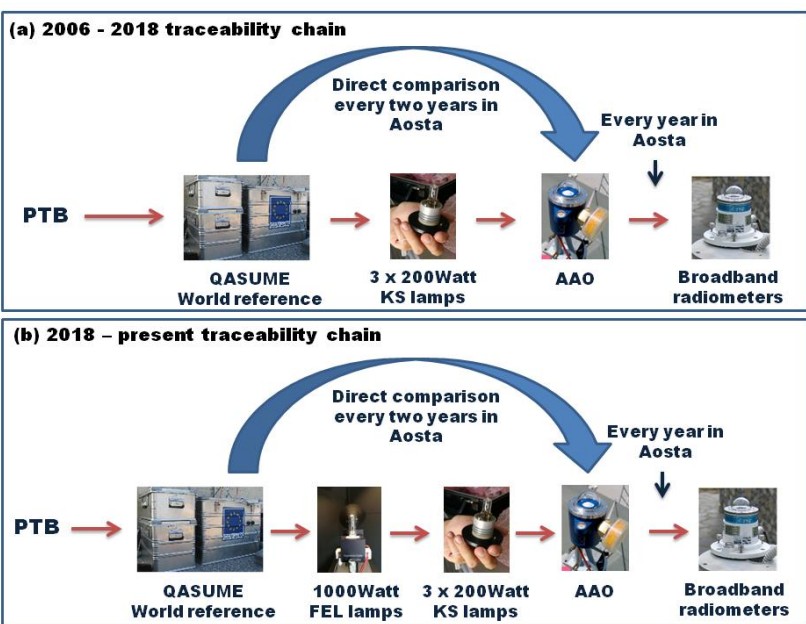

**Figure 5: Traceability chain for (a) 2006 – 2018 and (b) 2018 – present**

## 4.2 Re-evaluation and data versioning

We defined four available levels of measurements from the Bentham5541.

- **Level 0:** Raw photocurrent data
- **Level 1:** Near real-time irradiance spectra
- **Level 1.5:** Re-evaluated irradiance spectra
- **Level 2:** Re-evaluated irradiance spectra between two subsequent QASUME inter-comparisons

A more detailed description of each level of the data from AAO is provided in the following.

### 4.2.1 Level 0

The recorded signal during spectral scans of the solar (or lamp) irradiance is stored in ASCII files in units of nA. Before
being saved, the measurements have been corrected for the effect of dark signal and then have been reduced to an
amplification level of one. The dark signal, the level of amplification, and the temperature of the diffuser are also saved in
the same ASCII files. We consider the information in these ASCII files as the Level 0 data.

### 4.2.2 Level 1

Directly after measuring the irradiance from the 200 Watt lamps, the responsivity of the instrument is calculated from each
lamp following the methodology described in Bernhard and Seckmeyer (1999). If the difference between the average
calibration factors from the two lamps is larger than 1%, measurements are repeated, and the reason that caused the

difference is investigated. The lamp current is stabilized at 6.3 (± 0.0004) A by a feedback-loop controlled circuit using a precision resistor and voltage, both calibrated once a year. The voltage across the lamp is monitored and recorded during measurements. The spectral calibration factors calculated as the average responsivity at each wavelength from the acceptable
lamp scans are the Level 1 calibration factors which are thereafter automatically applied on spectral measurements. The recorded spectra are processed using the SHICrivm algorithm, and in near real time the information is stored in the database and after further processing (also performed in near real time) it is uploaded on the ARPA VdA web-page ([http://www.uv-index.vda.it](http://www.uv-index.vda.it)). After May 5[th] of 2017 all measurements are also automatically corrected for the effect of temperature on the transmissivity of the Teflon diffuser

The following information is produced by further processing the spectra and is then uploaded on the web-page:

- The UV-A irradiance (calculated by integrating measurements in the range 315 – 400 nm)
- The UV index (calculated by weighting measurements in the range 290 – 400 nm with the erythemal effective spectrum defined by the International Commission on Illumination (CIE) (CIE, 1999) and then integrating in the range 290 – 400 nm)
- Total ozone, calculated using a methodology similar to the methodology described in Bernhard et al. (2003).

On the web-page the particular quantities are directly compared with the corresponding quantities from other sources (measurements from collocated instruments, model and satellite estimates). The UV index is measured by Brewer#066 and the dual-band Kipp & Zonen radiometer, and the UV-A irradiance by the latter. Furthermore, on a daily basis, simulations of the clear-sky spectral irradiance in the range 290 – 400 nm are performed using the radiative transfer model UVSPEC of the
libRadtran package (Emde et al., 2016). For the model simulations, the atmosphere is considered to contain a constant amount of continental aerosol. The total ozone used as input for the model is the forecast provided by the German Meteorological Service (DWD) for that day. A more detailed description of the model settings is provided in Diémoz et al. (2013). The clear-sky UV index and the UV-A irradiance are again calculated from the model spectra. Finally, the UV index and the UV-A irradiance from the Bentham5541 and all the above sources are compared to each other. Total ozone from the
Bentham5541 is compared with the total ozone from the Brewer#066 and satellite retrievals from the Ozone Monitoring Instrument (OMI) on board Aura satellite (Levelt et al., 2018).

This way, both, the operators and the users of the web-page, can control the validity of the presented information. Large differences among the instruments and/or extremely high/low values relative to the modelled irradiances may imply operational problems of one of the instruments. On a daily basis unexpected results are investigated, and erroneous
measurements are directly removed from the database. The Level 1 dataset includes the spectra and the responsivity files described above.

### 4.2.3 Level 1.5

The Level 1 dataset is inhomogeneous. For example, although the diffuser temperature is recorded at the beginning of each scan, the spectra before May 5[th] of 2017 are not corrected for the effect of temperature on the transmittance of the Teflon

diffuser, while a correction has been applied thereafter. The level 1.5 dataset has been homogenized by post-correcting the whole Level 1 dataset for this effect. The changes are discussed in more detail below.

All spectra have been post-corrected for the effect of the diffuser temperature. In addition to the solar spectra, temperature correction has been applied on all lamp measurements (i.e. all calibration spectra). In Fig. 6, the ratio between the Level 1.5 and Level 1 irradiance at 305 nm and 495 nm is presented for the period 2010 – 2012, during which no other correction has 565 been applied to the measurements. The maximum differences resulting from the correction for the effect of temperature are in the order of 3% and the dependence on wavelength is small as can be perceived by Fig. 6. What is also obvious is that there is a clear annual cycle in the variability of the ratio. The effect of the diffuser temperature is more significant in winter and less significant in the summer.

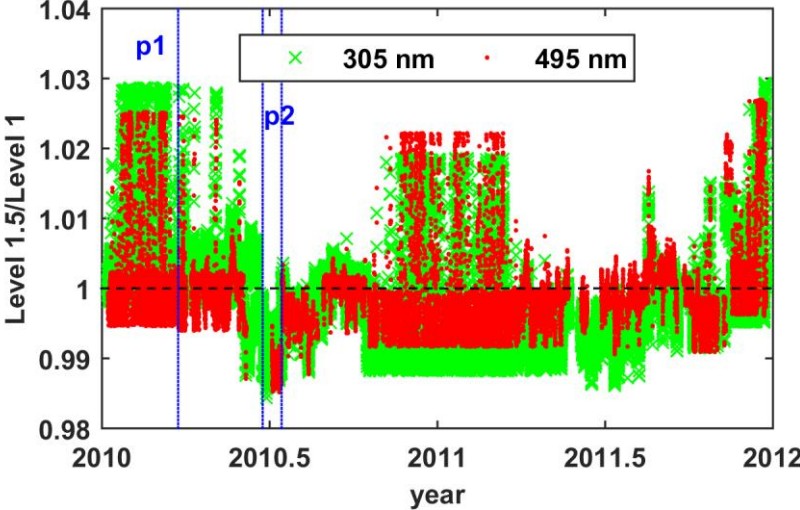

**Figure 6: Ratio between the Level 1.5 and the Level 1 irradiances, at 305 nm (green x) and 495 nm (red dots). Two different periods have been marked on the graph in order to assist further discussion: p1 at the beginning of 2010 and p2 at mid-2010. The two wavelengths refer to the average of a band, over ±5 nm around the central wavelength.**

The lowest values of the ratio at the beginning of 2010 (p1) are ~1.5% higher relative to the lowest values of the ratio in the period p2 (mid-2010 among blue dotted lines). This is because the temperature of the Teflon diffuser during the calibration 575 in the first case was between 30°C and 33°C. In the second case the temperature was between 36°C and 42°C. Thus, applying the correction for the effect of temperature resulted to a larger change for p2.

The very large ratios (~1.03) in p1 are due to the correspondingly large daily variability in the temperature of the diffuser (from ~15°C to ~32°C) in winter. The flat "top" in Fig. 6 corresponds to the minimum transmissivity of the diffuser (for the range of recorded temperatures) at 15°C. The lowest values of the ratio do not go below a certain limit, again because the 580 transmissivity (and subsequently the responsivity of the system) becomes maximal at 22°C. The difference of ~ 0.5% between the results for 305 nm and 495 nm is mainly because of the warming of the diffuser by the lamps during calibration resulting to temperature increase of 2-3°C from the beginning until the end of the scan. Since 2017 the temperature is

monitored with a step of 10 nm in the scan and the recorded values are used to correct the measurements. In order to take into account this effect for the period before 2017, an interpolation of the temperature values between consecutive scans is applied. When this is not possible, the value recorded at the beginning of the scan is used for the correction of all measurements in the scan.

### 4.2.4 Level 2

There are cases for which problems in the calibration procedure or the measurements have been detected a long time (of even months or years) after the data were stored in the database and uploaded on the web-page. Post-correcting the spectra for the above problems induces differences which in all cases are below 5%. Thus, we can consider that the Level 1 spectra can be safely used for informing the public without being misleading, and the Level 1.5 spectra are of good quality. However, changes in the order of 5% in the dataset may induce non-negligible differences or biases in the climatological analysis of the data. Thus, the whole calibration dataset has been recently re-evaluated on the basis of the current level of knowledge. The results of the inter-comparisons with QASUME have been used to certify that the applied corrections have improved the accuracy of the dataset. Thus, only spectra between consecutive QASUME inter-comparisons are classified as Level 2 spectra.

In Fig. 7, the ratio between the Level 2 and the Level 1 calibration factors at 310 nm, 400 nm, and 490 nm is presented for the period 2006 – 2019.

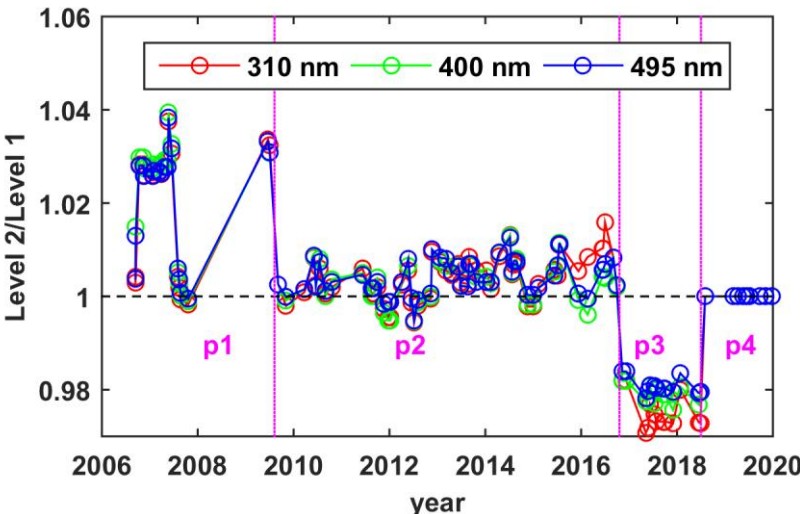

**Figure 7: Ratio between the Level 2 and Level 1 calibration factors. Four different periods have been marked on the graph in order to assist further discussion: p1 from July 2006 until July 2009, p2 from July 2009 until October 2016, p3 from October 2016 until June 2018, and p4 from June 2018 to 2020.**

In p2 the only difference between the Level 2 and the Level 1 calibration factors, is the correction of the former for the effect of temperature, which also means that there is no difference between the Level 1.5 and Level 2 spectra for the particular

period. In most cases the temperature during calibration was above the reference temperature of 28°C (to which both, the spectra and the calibration factors are interpolated) and the correction increased the calibration factors (representing the responsivity). Even in extreme cases the increase does not exceed 2%. Larger differences in the order of 3 – 4% are evident in p1 and p3. More specifically, the Level 2 calibration factors are in many cases higher by up to 3 – 4% in p1 and lower by 2 – 3% in p3. It is obvious that not correcting the spectra would introduce a positive bias in the order of 2 – 3% if trends

were calculated for the whole period. Analytical explanation of the causes of the differences in the used calibration factors is provided below.

During p1 new 200 Watt lamps were purchased by Schreder CMS, together with their calibration certificates. At some point however they were re-calibrated by PMOD-WRC. In the particular period the PFC was not brought to Davos together with the lamps for calibration. Thus, lamps were calibrated at Davos with a PFC which was different from the one used for the

calibration of Bentham5541 at AAO. In 2009 it was found that using the PFC of PMOD – WRC instead of that of AAO resulted to ~2.3% more irradiance reaching the sensor, most likely because small differences in the geometry of the two PFCs. Thus, the irradiances in the lamp certificates provided by PMOD – WRC were overestimated by ~2.3%, and subsequently the calibration factors calculated using the particular certificates were underestimated by the same amount. These errors affecting the Level1 and Level1.5 calibration factors (and spectra) have been corrected in the Level 2 dataset.

The irradiance of the lamps calibrated at Davos in 2006 was additionally overestimated by ~1% due to the then unknown reference plane of the QASUME diffuser (Gröbner and Blumthaler, 2007) . Thus, the irradiances of the lamps for the period 2006 – 2007 were additionally lowered by 1% (the calibration factors have been increased by 1%). The overall increase in the Level 2 (relative to the Level 1 and Level 1.5) calibration factors for September 2006 – August 2007 is up to ~3.3% (without taking into account the effect of temperature). It should be noted that at the end of 2007 the magnetothermic switch

of the Bentham5541 was broken and the system was stopped for several months. Thus measurements are not available during the entire year in 2008.

Replacement of the fore-optics of QASUME in 2016 does not allow the calibration of the 200 Watt lamps at Davos anymore. The last calibration of 200 Watt lamps at the facilities of PMOD – WRC was thus performed in 2016. Results of the 2017 inter-comparison with QASUME and the inter-comparison of broad-band radiometers held in Davos in 2018

(Hülsen et al., 2020) , as well as characterization in the new optical laboratory at AAO in 2018 confirm that the real irradiances are possibly 2- 4% higher than those measured in 2016. A possible explanation could be that something changed in the PFC (e.g. during shipment of the PFC in 2016 from Davos to Aosta, or after disassembling and reassembling the PFC). Thus, all calibration factors used during p3 have been corrected (decreased by ~2.5%) based on the results of the characterization of the 200 Watt lamps in the new optical laboratory at AAO in 2018. The accuracy of the new irradiances

(calculated in mid 2018) was also confirmed during the 2019 inter-comparison with QASUME.

Since July 2018 (p4) no other changes have been applied on the used calibration factors and the ratio is constantly 1.

Summarizing, the Level 2 dataset is corrected for the effect of temperature of the diffuser, and homogenized taking into account calibration problems and changes in the calibration scale. The latter is the main difference with the Level 1.5

spectra. The high quality of the Level 2 spectra has to be assured by consecutive comparisons with QASUME (i.e. only

spectra between consecutive QASUME inter-comparisons can be Level 2). The level 2 (as well as the Level 1 and Level 1.5) calibration factors are used since the day they are measured or since the day of a known change in the instrument, and no linear interpolation has been applied between consecutive calibrations. Slow changes in responsivity between consecutive calibrations are taken into account in the overall uncertainty of the final spectra.

### 4.2.5 Data storage

Each Level 1.5 and Level 2 spectrum is saved in the database in the form of a NetCDF file together with all metadata which are necessary for the production of the Level 2 spectra from the raw data (i.e. dark signal, diffuser temperature, spectra before the correction for wavelength shifts, spectral responsivity etc), as well as the estimated uncertainty at each wavelength (Sect. 5). The NetCDF file is structured according to the version 1.8 of the Climate and Forecast (CF) metadata conventions (Eaton et al., 2020). In the same file the following effective doses which have been calculated from the particular spectrum

are also stored:

- Erythemal irradiance (CIE, 1999)
- Effective dose for the production of vitamin D (Bouillon et al., 2006)
- Effective dose for DNA damage (Setlow, 1974)
- Effective dose for plant damage (Caldwell, 1971)
- Effective dose for plant growth (Caldwell, 1971)
- Integrals of UV-B, UV-A and total UV irradiance

### 4.2.6 Inter-comparisons with QASUME

Analytical results of the inter-comparisons with QASUME between 2006 and 2019 are available on the web-page of PMOD-WRC (https://www.pmodwrc.ch/wcc_uv/qasume_audit/reports/). The only exception is the 2014 inter-comparison held in

Davos for which the results are not publicly available. In Fig. 8, the average ratio between the measurements of the two instruments at 310 nm (± 5 nm average), 390 nm (± 5 nm average), and 490 nm (± 5 nm average), as well as the corresponding intervals of the 5/95 percentiles are presented. Shaded grey areas represent the combined expanded uncertainty of the Bentham5541 Level 2 (see Sect. 5) and the QASUME spectra for each intercomparison. The Bentham5541 uncertainties are reported and analytically discussed in Sect. 5. The two fold uncertainties of the spectra

measured by QASUME during intercomparisons are: 4.6% for 2006 – 2013, 2.9% for 2014 – 2017, and 1.9% for 2019 (Hülsen et al., 2020;Hülsen et al., 2016). These numbers, as well as the uncertainties of Bentham5541 which have been used in order to calculate the combined uncertainty are for SZAs below 75° and wavelengths longer than 310 nm. The uncertainties of the two instruments are not completely independent since the same irradiance reference is used for the calibration of both. Thus, the uncertainty in the calibration of the reference 1000 Watt lamps (used by PMOD – WRC to

transfer the calibration to the working standard lamps of both institutes) has not been taken into account when the overall radiometric uncertainty of Bentham5541 was calculated. The presented ratios are for all SZAs. Both, the original (between the Level 1 and the QASUME spectra) and the re-evaluated (between the Level 2 and the QASUME spectra) ratios are presented. It should be noted at this point that in some cases the results in the PMOD – WRC reports differ from those for Level 1 spectra because a re-evaluation was performed directly after particular campaigns.

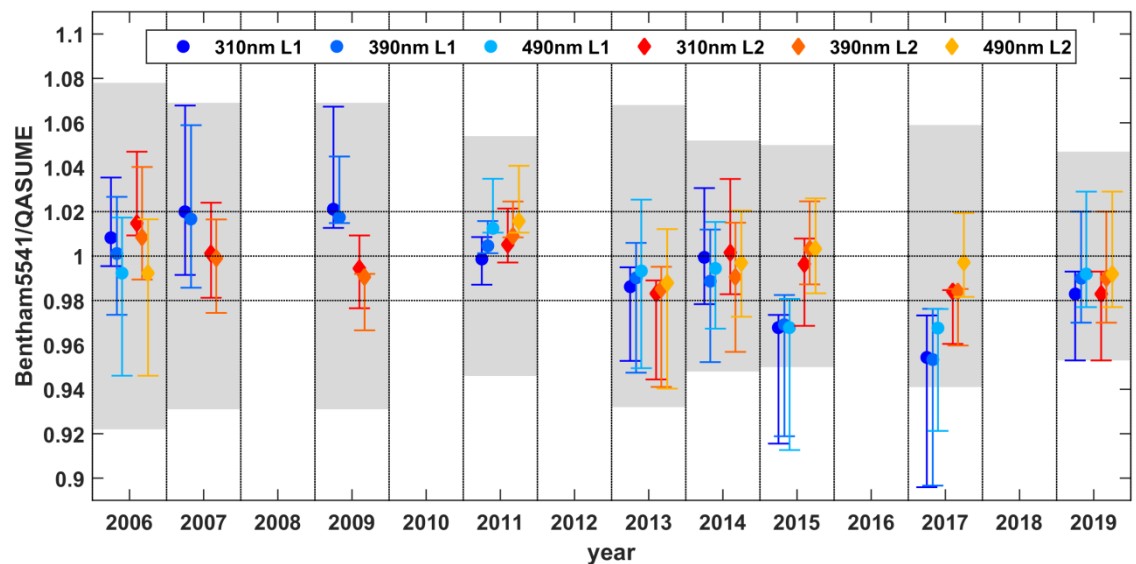


**Figure 8: Average ratio between the measurements of Bentham5541 and QASUME at 310 nm, 390 nm, and 490 nm, and the corresponding 5/95 percentiles Shaded grey areas represent the combined expanded uncertainty of the Bentham5541 Level 2 and the QASUME spectra for each intercomparison.**

The general conclusions coming from Fig. 8 can be summarized as follows:

- For most years there is a clear improvement of the results when the Level 2 dataset is used. The average ratio between Level 1 Bentham5541 and QASUME spectra ranges from -4% to +2%, while the average ratio between the Level 2 Bentham5541 and the QASUME spectra is always within ±2%.
- In all cases, the limits of the 5/95 percentiles for the Level 2 to QASUME ratio are within ±6%. This is because of the relatively larger differences between the two instruments for SZAs between 80° and 90°. If we exclude SZAs
above 75° from the analysis, then the limits are within ±4%. However, in both cases they are within the limits of the two fold combined expanded uncertainties.
- The 5/95 percentile intervals become narrower in some cases (e.g. 2007, 2009, 2015, 2019) for the Level 2 spectra. This is either because the used calibration factor changed for some of the days of the campaign (i.e. the same responsivity is used for the whole period of the campaign instead of different responsivities for different days), or
because problematic spectra were removed from the dataset.
- With the exception of 2006, the dependence of the ratio from wavelength is small (less than 1%) for both, Level 1 and Level 2.

The remaining differences between the Level 2 and QASUME spectra have been in all cases explained; this way we were able to further quantify the effect of the factors that cause them on the Bentham5541 measurements.

- The dependence of the ratio from wavelength, as well as the large variability in **2006**, are due to the imperfect positioning of the fiber in the OH which induced a dependence of the ratio from the position of the sun.
- The differences between the Level 1 and Level 2 ratios in **2007** and **2009** are due to the changes in the used lamp irradiances, described in Sect. 4.2.4.
- The average difference of ~2% between Bentham5541 and QASUME in **2013** is again because of the non-optimal positioning of the fiber in the OH. The position of the fiber optics in the OH was optimized in 2007 and 2014, after detecting problems in 2006 and 2013 respectively.
- The **2014** inter-comparison took place in Davos and not in Aosta. As already discussed, small mis-leveling of the diffuser in the particular period affected the measurements and resulted to increased variability in the ratio (relative to the campaigns in 2013 and 2015).
- The average difference of ~2% between Bentham5541 and QASUME in **2017** and **2019** is because the fore-optics of QASUME was upgraded in 2016, and since then its angular response is better than the angular response of Bentham5541. A diffuser with improved angular characteristics will replace the one used by Bentham5541 in order to solve this issue.

In all cases the detected problems have been taken into account in the calculation of the overall uncertainty budget. A
comprehensive discussion about the calculation of the overall uncertainty budget for different periods is provided in the following section.

## 5 Uncertainties

Diémoz et al. (2011) have already estimated the overall uncertainty in the Level 1 spectral measurements of the Bentham5541 based on the methods described in Gröbner and Sperfeld (2005) , Bernhard and Seckmeyer (1999) , and the
Guide to the expression of Uncertainty in Measurement  (BIPM et al., 2008). In this study the uncertainties reported by Diémoz et al. (2011) have been updated for the Level 2 spectra, as well as integrated quantities such as the UV-A and the erythemal irradiance. The overall uncertainty budget is not identical for all years since different problems affect measurements in different sub-periods.  In the following, the word uncertainty denotes the standard (one fold) uncertainty unless something else is specified.

**5.1 Radiometric uncertainty**

Uncertainties in the calibration procedure play a very important role in the overall uncertainty budget. The main factors that are responsible for these uncertainties are discussed in the following.

### 5.1.1 Lamp certificate

When a lamp is characterized in an accredited laboratory, an uncertainty budget is specified for the lamp irradiance at each wavelength. The uncertainty budget generally depends on the individual characteristics of the lamp and the characterization procedure. The standard uncertainty reported in the lamp certificates until 2010 was 2.3% for all wavelengths. Since 2011 the reported uncertainties have been smaller and wavelength dependent. The uncertainty reported in the certificates of the 200 Watt and 1000 Watt lamps between 2011 and 2019 decreases with wavelength, ranging from 0.9 – 1.4% at 300 nm to 0.5 – 0.7% at 500 nm. When the lamps which have been calibrated at the accredited laboratory return to AAO, they are used as references in order to recalibrate other lamps. An uncertainty budget has been also calculated for the lamps which are calibrated at the AAO.

### 5.1.2 Calibration transfer

The uncertainty due to the transfer of the calibration from the reference to the working standard lamps has not been taken into account by Diémoz et al. (2011). In the study of Kazadzis et al. (2005) the uncertainty due to the transfer of the calibration from the reference to the working standard lamps for a calibration scheme similar to that of AAO has been estimated to ±2%. In our case, a significant contribution to the calibration transfer uncertainties comes from the statistical noise in the measurements which is different in different periods. Before 2018, the additional uncertainty in the Level 2 calibration factors due to the calibration transfer is 0.4% for all wavelengths. After using the new calibration system in 2018 (200 Watt lamp PFC with the extender and 1000 Watt lamps in the laboratory) the uncertainty increases to 0.7% and 0.6% at 300 nm and above 310 nm respectively.

### 5.1.3 Instability

The instability of the instrument can be attributed to a number of factors (some of which have been discussed in Sect. 3.7). In the period between two consecutive calibrations, the Level 2 calibration factors do not change by more than 1.5%, unless something in the system setup has been changed. Assuming a rectangular probability we estimate that the relative uncertainty is 0.4%, which is in agreement with the results of Diémoz et al. (2011).

### 5.1.4 Heating of the diffuser

All experiments described in Sect. 3.2 suggest that the remaining error after correcting measurements for the effect of temperature is less than 0.5%. Furthermore, the temperature of the whole surface of the Teflon diffuser is homogeneous within 2°C during lamp measurements which cannot justify errors larger than 0.2%. Assuming again a rectangular probability we estimate that the relative standard uncertainty is equal to 0.2%.

### 5.1.5 Non linearity and changes in responsivity after exposure to high radiation levels

There is no detectable change in the responsivity of Bentham5541 when it is exposed to high radiation levels that may occur during clear skies in summer months. There is also no sign of non linearity in the measurements (as already explained in Sect. 3.5). Nevertheless, even if there is some uncertainty related to these phenomena it is set to zero because it is part of the uncertainty from instrument instability (Sect. 5.1.3).

### 5.1.6 Lamp aging

Analysis of the record of the lamp irradiances showed that the 200 Watt lamps are stable within ±1% over the years, and after a certain point they begin drifting. The drift was never found to be larger than 2% for the period of one year between two consecutive calibrations of each lamp, either at the facilities of PMOD-WRC, or at AAO. Based on the above we estimate that the relative standard uncertainty is again the same with that reported in Diémoz et al. (2011), i.e. 0.5%.

### 5.1.7 Lamp current and wavelength stability

As has been discussed in Gröbner and Sperfeld (2005), instabilities in the lamp current and the wavelength scale of the instrument may also affect the calibration. The corresponding standard uncertainties are in both cases estimated to 0.1%.

### 5.1.8 Statistical noise

For the period before 2018 the uncertainty due to the statistical noise of the measurements is that reported in Diémoz et al. (2011), i.e. ±0.2% at 300 nm and 0.1% at longer wavelengths. However, since July 2018, 1000 Watt FEL lamps are used as reference for the calibration of the 200 Watt lamps at AAO. The signal of the 1000 Watt lamps is lower relative to the signal of the 200 Watt lamps with the short calibrator setup resulting to higher noise. Furthermore, the distance between the sensor and the 200 Watt lamps during calibration has been increased (the long calibrator setup is currently used) resulting again to lower signal, and more noise. Thus, the corresponding uncertainties are estimated to 0.6% and 0.4%.

### 5.1.9 Other sources of uncertainty

The results of the 2017 inter-comparison with QASUME, as well as characterization in the new optical laboratory in 2018 clearly showed that the lamp irradiances used since 2016 were not accurate and had to change. As discussed in Sect. 4.2.4 we estimated that the problem started when the lamps returned from Davos in 2016. Although the correction has been applied from the specific time point and on, it was not possible to independently prove that the problem did not start earlier or later. Thus, it is still possible that the calibration factor may be systematically over- or under-estimated in part of the period between the inter-comparisons of 2015 and 2017. Additional uncertainty of 1.6% has been added to the overall uncertainty of the calibration factor for the particular period.

### 5.1.10 Reproducibility of the calibration setup

When the PFC short setup is used for the calibration, the distance between the lamp and the diffuser is ~ 12 cm. Thus, even small changes in the distance between the lamp and the diffuser result to significant changes in the measured irradiance. Investigation of the record of the 200 Watt lamp measurements showed that the difference between close time scans where the OH was pulled out and then placed back into the PFC between the scans did not exceed 0.5%. In order to further investigate the reproducibility of the distance, five scans of the spectrum from a 200 Watt lamp were performed within a few

tenths of minutes. The OH was pulled out and then placed back between consecutive scans. Again, the maximum average difference between different scans was 0.5%. The difference of 0.5% in the measured irradiance corresponds in a reproducibility of 0.3 mm in the distance.

    When the PFC long setup is used the distance between the lamp and the diffuser is ~ 30 cm. In this case, a difference of 0.3 mm in the distance between the OH and the lamp results to a difference of 0.2% in the measured irradiance. As with the

short setup, consecutive scans of the spectrum from a 200 Watt lamp were performed which resulted to average differences which were in all cases within the noise of the measurements (i.e. smaller than 0.2%). The same test using the setup for the 1000 Watt lamp measurements also resulted to non detectable differences.

    In order to investigate if a rotation of the diffuser of a few degrees around its axis affects the results of the calibration, measurements of the irradiance from the 200 and 1000 Watt lamps were performed for slightly different positions of the

diffuser (rotation up to ~ 30° around its axis, clockwise and anticlockwise). No differences were detected between measurements at different rotation angles. Summarizing, when the short setup of the PFC is used the estimated uncertainty related to the reproducibility of the calibration setup is 0.2%. When the long setup of the PFC or the setup for the 1000 Watt lamp measurements are used the corresponding uncertainties are negligible.

### 5.1.11 Overall radiometric uncertainties

Based on the above discussion we consider that the uncertainty in the irradiance of the lamps used for the calibration differs significantly between the periods 24/07/2006 – 17/04/2011 and 18/04/2011 - present. During the former period the standard uncertainty is considered 2.3% for all wavelengths. In the latter period the uncertainty depends on lamp and wavelength. Thus, a rigorous calculation of the uncertainty would require taking into account the uncertainties of the lamps used for each calibration and consider different uncertainty budget for each period between consecutive calibrations. However, for

simplicity we consider that for the particular period the uncertainty in the lamps' irradiance is equal to the maximum reported uncertainty in all certificates and wavelength depended (decreasing from 1.4% at 300 nm to 0.7% at 500 nm). This way we may slightly overestimate the overall radiometric uncertainty for particular periods (by up to 0.5% at 300 nm and 0.2% at 500 nm). The results of the analysis discussed in this section are summarized in Table 1, where the overall radiometric uncertainty for different periods is presented.


**Table 1: Contribution of different sources in the overall radiometric uncertainty (in %) o the Level 2 AAO spectra.**

| Period | 24/07/2006 – 17/04/2011 | | | 18/04/2011 – 16/06/2015 | | | 17/06/2015 – 24/09/2017 | | | 25/09/2017 – 17/07/2018 | | | 18/07/2018 – present | | |
|---|---|---|---|---|---|---|---|---|---|---|---|---|---|---|---|
| Wavelength (nm) contribution | 300 | 310 - 400 | 400 - 500 | 300 | 310 - 400 | 400 - 500 | 300 | 310 - 400 | 400 - 500 | 300 | 310 - 400 | 400 - 500 | 300 | 310 - 400 | 400 - 500 |
| Lamp irradiance | 2.3 | 2.3 | 2.3 | 1.4 | 1.1 | 0.7 | 1.4 | 1.1 | 0.7 | 1.4 | 1.1 | 0.7 | 1.4 | 1.1 | 0.7 |
| Calibration transfer | 0.4 | 0.4 | 0.4 | 0.4 | 0.4 | 0.4 | 0.4 | 0.4 | 0.4 | 0.4 | 0.4 | 0.4 | 0.7 | 0.6 | 0.6 |
| Instability | 0.4 | 0.4 | 0.4 | 0.4 | 0.4 | 0.4 | 0.4 | 0.4 | 0.4 | 0.4 | 0.4 | 0.4 | 0.4 | 0.4 | 0.4 |
| Heating of the diffuser | 0.2 | 0.2 | 0.2 | 0.2 | 0.2 | 0.2 | 0.2 | 0.2 | 0.2 | 0.2 | 0.2 | 0.2 | 0.2 | 0.2 | 0.2 |
| Lamp aging | 0.5 | 0.5 | 0.5 | 0.5 | 0.5 | 0.5 | 0.5 | 0.5 | 0.5 | 0.5 | 0.5 | 0.5 | 0.5 | 0.5 | 0.5 |
| Lamp current | 0.1 | 0.1 | 0.1 | 0.1 | 0.1 | 0.1 | 0.1 | 0.1 | 0.1 | 0.1 | 0.1 | 0.1 | 0.1 | 0.1 | 0.1 |
| Wavelength stability | 0.1 | 0.1 | 0.1 | 0.1 | 0.1 | 0.1 | 0.1 | 0.1 | 0.1 | 0.1 | 0.1 | 0.1 | 0.1 | 0.1 | 0.1 |
| Statistical noise | 0.2 | 0.1 | 0.1 | 0.2 | 0.1 | 0.1 | 0.2 | 0.1 | 0.1 | 0.2 | 0.1 | 0.1 | 0.6 | 0.4 | 0.4 |
| Calibration setup reproducibility[1] | 0.2 | 0.2 | 0.2 | 0.2 | 0.2 | 0.2 | 0.2 | 0.2 | 0.2 | 0.2 | 0.2 | 0.2 | <0.1 | <0.1 | <0.1 |
| Standard uncertainty | 2.4 | 2.4 | 2.4 | 1.6 | 1.4 | 1.1 | 2.3 | 2.1 | 1.9 | 1.6 | 1.4 | 1.1 | 1.8 | 1.5 | 1.2 |
| Expanded uncertainty | **4.9** | **4.9** | **4.9** | **3.2** | **2.7** | **2.1** | **4.6** | **4.2** | **3.8** | **3.2** | **2.7** | **2.1** | **3.6** | **3.0** | **2.4** |

## 5.2 Overall measurement uncertainties

Although radiometric uncertainties play a major role in the formulation of the overall uncertainty budget, some more uncertainty sources in actual solar measurements also have a significant contribution. The most important are discussed in the following sections.

### 5.2.1 Diffuser temperature

The uncertainty calculated in Sect. 5.1.4 is also representative for the field measurements. Thus, the standard uncertainty in the field spectral measurements is also 0.2%.

### 5.2.2 Changes in responsivity and linearity

Since 24 July 2006, the relative uncertainty is negligible as discussed in Sect. 5.1.5.

---

[1] For the period 18/07/2017 – present, the uncertainty related to the reproducibility of the calibration setup has not been taken into account in the calculation of the overall uncertainty budget.

### 5.2.3 Cosine and azimuth response

Based on the results of Sect. 3.2 we estimate the following standard uncertainties due to the imperfect cosine response of the instrument.

For SZA below 70°: 0.3% in the UV-B, 0.6% in the UV-A, and 0.7% in the VIS

For SZA above 70°: 0.1% in the UV-B, 0.6% in the UV-A, and 1.2% in the VIS

These uncertainties correspond to Aosta altitude and atmospheric conditions for the Level 2 products.

Inhomogeneities in the azimuth response of the diffuser generally have a negligible contribution in the uncertainty budget of the measurements. As discussed in Sect. 3.3, there are however periods, during which mis-levelling of the diffuser or improper positioning of the fiber optic in the OH induce more significant uncertainties. In these periods, the relative standard uncertainty may be up to 2%.

### 5.2.4 Other uncertainty sources

Other sources of uncertainty which have been already discussed in previous studies are the statistical noise and the wavelength misalignment of the instrument. Both are more significant for short wavelengths and large SZAs (4.6% and 2.4% respectively at 300 nm for SZA=75°) and less significant for larger wavelengths and smaller SZAs (below 0.3% and 0.9% respectively for wavelengths longer than 400 nm and SZA below 50°). The contribution of these sources has been already discussed in Diémoz et al. (2011).

### 5.2.5 Overall uncertainty budget

The overall uncertainty budget for different periods has been calculated by taking into account the same factors as Diémoz et al. (2011). The results for different periods are presented in Table 2.

**Table 2. Expanded (k=2) uncertainty (in %) in the spectral measurements of the Bentham5541 for different periods. The series for Level 2 spectra stops in July 2019 when the last QASUME intercomparison took place.**

| Period | 300nm SZA<50° | 300nm SZA<75° | 310–400nm SZA<50° | 310–400nm SZA<75° | 400–500nm SZA<50° | 400–500nm SZA<75° |
|---|---|---|---|---|---|---|
| 24/07/2006–20/12/2006 | 7.8 | 12.2 | 6.6 | 7.0 | 6.0 | 6.4 |
| 21/12/2006 – 17/04/2011 | 6.5 | 11.5 | 5.4 | 5.6 | 5.4 | 6.0 |
| 18/04/2011 – 26/08/2011 | 5.6 | 10.8 | 3.6 | 4.1 | 3.4 | 4.2 |
| 27/08/2011 – 07/07/2014 | 6.0 | 11.0 | 4.2 | 4.6 | 5.2 | 5.8 |
| 08/07/2014 – 13/01/2015 | 5.6 | 10.8 | 3.8 | 4.2 | 3.4 | 4.3 |
| 14/01/2015 – 16/06/2015 | 5.7 | 10.8 | 3.6 | 4.0 | 3.4 | 4.2 |

| | | | | | | |
|---|---|---|---|---|---|---|
| **17/06/2015 – 24/09/2017** | 6.6 | 11.4 | 4.8 | 5.2 | 4.6 | 5.2 |
| **25/09/2017 – 17/07/2018** | 5.6 | 10.7 | 3.6 | 4 | 3.4 | 4.2 |
| **18/07/2018 – 08/07/2019** | 5.7 | 10.9 | 3.8 | 4.2 | 3.4 | 4.3 |

It can be perceived from Table 2 that for the full time series and for SZAs below 75° the expanded uncertainty at 300 nm is 12% or smaller, while for wavelengths in the UV-A and VIS it is below 7%.

A summary of the main factors contributing to the overall uncertainty for different wavelengths and SZAs is provided in Table 3.

**Table 3. Main factors contributing to the overall uncertainty budget.**

| Contribution | Description and relative standard (k=1) uncertainty |
|---|---|
| Radiometric uncertainty | Main contributor to the overall uncertainty. Ranges from 2.4% to 1.1% depending on period and wavelength. |
| Diffuser temperature | After correcting measurements for this effect it is 0.2%. |
| Angular response | Depends on wavelength, period and SZA. Under usual conditions, for SZA<70° increases from 0.3% to 0.7% for wavelengths 300 – 500 nm. For SZA>70° the corresponding range is 0.1 – 1.2%. Mis-levelling of the diffuser or improper positioning of the fiber optic result to uncertainties up to 2%. |
| Changes in responsivity after exposure to high signal and linearity | The responsivity of the system has not been found to change after exposure to high signal levels (after 24 July 2006) under the usual operational conditions. Relative uncertainty has been set to 0%. |
| Instability | Estimated to 0.4% for the whole period of study. |
| Statistical noise | Mostly affects the shorter wavelengths at high SZA. For SZA<50° it is 0.3% for wavelengths above 310 nm and 0.8% at 300 nm. For SZA=75° the corresponding numbers are 0.9% and 4.6%. |
| Wavelength misalignment | Mostly affects the shorter wavelengths. From 2.1% to 2.4% at 300 nm for SZAs 50° - 75° respectively. About 0.9% for wavelengths longer than 310 nm. |

The uncertainties in the erythemal and total UV-A irradiances (calculated from the spectra measured by the AAO) have been also quantified, assuming that uncertainties due to wavelength misalignment are negligible. For SZA below 75° the expanded uncertainty in erythemal irradiance is in the order of 5% before 17/04/2011 and 4% thereafter. The corresponding uncertainties in UV-A are 3% and 2% respectively.

## 6 Summary and future prospects

The network of the Aosta Valley has been the first UV monitoring network in Italy. The reference instrument of the network is a Bentham DTMc300 spectroradiometer which has been performing automated continuous spectral scans of the solar irradiance in the range of 290 – 500 nm since 2006, and is used as the reference instrument for all broadband instruments of the regional network of the Aosta Valley. A rigorous QA/QC protocol and a strong traceability chain ensure the good quality

of the spectra recorded by the Bentham5541. In the present study the methodology used to characterize the instrument for its individual characteristics, correct the measurements, and quantify the relative uncertainties is described. Recently, the whole dataset has been re-evaluated and a new, highly accurate UV dataset has been produced.

In addition to the re-evaluation of the whole record of calibration factors, the accuracy of the Level 2 spectra has been improved significantly relative to the Level 1 spectra after applying a correction for the effect of temperature on the
transmissivity of the Teflon diffuser. The accuracy of the correction for this effect has been confirmed by different experiments. It has also been shown that the responsivity of the instrument does not change, even after exposure to very high signal levels since 2006 when the HV of the PMT was set to a very low level (~ 400V) relative to the HV of other similar instruments (usually above 600V). The decrease in HV led however to a large decrease in the responsivity, thus a lower signal to noise ratio. In the near future there will be an effort to determine an optimal level of the HV for which the signal to
noise ratio will increase without introducing non-linearity.

Uncertainties related to the calibration of the instrument contribute significantly to the overall uncertainty budget. Improved characterization (by PMOD – WRC) of the lamps used for the calibration of the AAO after 2011 led to more accurate determination of their irradiances and a reduction of the overall standard uncertainty in the measurements by ~1%. For wavelengths above 310 nm and SZAs below 75° the expanded uncertainty in the former period is ~6%, while in the latter it
is ~4%. At 300 nm the expanded uncertainties are 10 – 12% and are dominated by uncertainties due the statistical noise and wavelength instability. The overall uncertainty budget varies through the years, mainly as a result of different problems that affected the measurements during different periods. Further reduction in the overall uncertainty budget is expected in the future due to the planned replacement of the currently used Teflon diffuser by a new diffuser with improved angular response. This study clearly points out the necessity of a maintaining a strong traceability chain to a reference instruments, in
addition to keeping a strict QA/QC protocol. This way it is possible to detect dis-continuities and errors in the time-series which might induce significant biases in the study of the trends of the spectral solar UV irradiance.

The new, low uncertainty Level 2 dataset is suitable for climatological studies and validation UV retrievals from satellite measurements and models. The time-series of the noon UV index as it has been calculated from the spectra is presented in Figure 9.

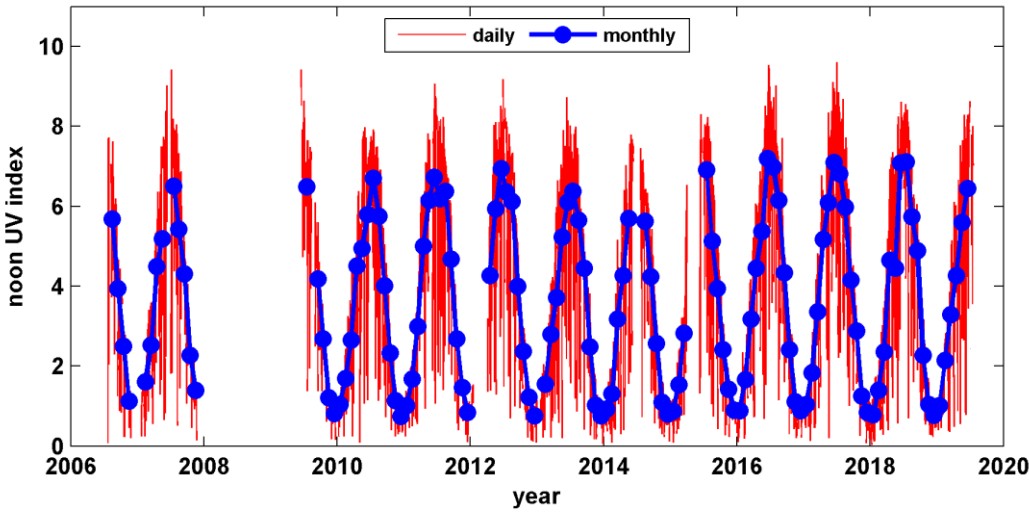


**Figure 9: The daily and monthly average noon UV index for the period 2006 – 2019, for which Level 2 data are available.**

The noon UV index for each day has been calculated as the average of available measurements for ±15 minutes around the exact local noon. The monthly averages have been calculated when the noon-time UV index is available for at least 15 days per month. The UV index in Aosta-Saint Christophe ranges from very low values (near zero) in winter and under cloudy

conditions to very high values of ~9 in summer. Monthly averages range from 1 in winter to 6 or 7 in summer months. The large variability of the daily and monthly UV index is indicative for the need of continuous UV monitoring. Further climatological analysis is however out of the scope of the present study and will be addressed in an upcoming article.

## 7 Data availability

The new Level 2 spectral UV dataset for the period 2006 – 2019 is freely available at

https://doi.org/10.5281/zenodo.4028907 (Fountoulakis et al., 2020). The noon UV index used to create Fig. 9 is also available at the same repository. Analytical information regarding the format and the context of the provided files can be found in the accompanying readme.txt file. The provided files are in csv format and include the spectra as well as the time of the beginning and the end of each scan. The NetCDF stored in the database (see Sect. 4.2.5) have not been directly uploaded since they contain a very large amount of information which is not useful to the end user. However they are freely available

and can be provided by the station PI (Henri Diémoz, e-mail: h.diemoz@arpa.vda.it). In case of any publication involving the particular dataset, co-authorship should be considered if the dataset plays a substantial role in the study. Acknowledgement of the dataset should be in all cases determined in consultation with the station PI. It is planned that the Level 2 spectra will also be submitted to the WOUDC (https://woudc.org/) and the European UV database (EUVDB) (http://uv.fmi.fi/uvdb/) (Heikkilä et al., 2016) in due course.

## Authors contribution

IF has re-evaluated the UV dataset of Bentham5541 and prepared the original draft of the manuscript. He has also contributed to the instrument characterization and calibration. HD has developed the UV monitoring Network of the Aosta Valley and is currently the PI and responsible for the maintenance, characterization and calibration of all UV monitoring instruments of network, including the reference Bentham spectroradiometer. GH, AMS, and JG have assisted the characterization and the maintenance of the Bentham spectroradiometer, and have provided useful advises throughout the period 2006 – 2019 which helped to improve the quality of measurements. All authors contributed to the preparation of the final version of the manuscript.

## Competing interests

The authors declare that they have no conflict of interest.

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
