# Peer review of "Monitoring of solar spectral ultraviolet irradiance in Aosta, Italy"

_Earth System Science Data, 2020_

## Referee Comment (RC1) · Anonymous Referee #1 · 12 Aug 2020

General remarks:

The manuscript by Fountoulakis et al. describes a new dataset of UV radiation, measured by a Bentham spectroradiometer in northern Italy. The paper includes descriptions of the instrument's characterization, corrections applied to the measurements, the traceability of calibrations, the data's uncertainty, comparisons with the QASUME reference spectroradiometer, and available data products. The good agreement between measurements of QASUME and the author's instrument gives confidence in the good quality of data from the latter. While data of this instrument have been published before, the manuscript focuses on a new data version ("Level 2") and its uncertainty budget. The work is therefore appropriate for the journal of Earth System Science Data. Considering that spectral UV data with a well documented uncertainty are sparse, the new

dataset is of value for the scientific community. I downloaded one of the datasets from the website indicated in the paper (https://doi.org/10.5281/zenodo.3934324) and compared a few spectra with my own model calculations. I found the data easy to use and did not encounter a problem.

Methods and materials are generally described in sufficient detail in the manuscript, however, there are many minor issues (see Specific Remarks below) that need to be addressed before the manuscript can be published. In general, when referring to an "expanded uncertainty", please also provide the coverage factor. It is likely k=2, corresponding to a confidence level of 95%, but this is often not clear.

The paper is well structured but in need for extensive copy-editing. While the level of English is sufficient to understand the text, there are numerous grammatical errors (e.g., the placement of adverbs), which should be corrected before publication. In my review, I only point out language issues that could be missed by the copy editor. I trust that the remaining errors will be fixed during the production stage. The formatting of Table 1 and 2 is unusual. The format should also be corrected during production.

Specific remarks:

L22: "expanded uncertainty": provide the coverage factor (likely k = 2)

Figure 1: It would be much better to show a topographical map instead of a political map.

L174: "As already discussed the Bentham5541 is traceable to world reference QA-SUME." I would say the measurements are traceable to the scale of spectral irradiance established by PTB. At least the chain shown in Figure 5 starts with PTB. QASUME is only used for QA/QC, but data are not scaled to match QASUME's measurements. The role of QASUME should be better described.

Section 3.1: either use the term "dark current" or "dark signal" but not a mixture of both.

L195: Why would the dark current depend on "the level of the intensity of the incoming

light"? The dark current is presumably measured with a physical shutter closed, or perhaps determined by scanning below 280 nm where no solar radiation can be detected at the Earth's surface (presuming that there is no stray light). So the "level of the intensity of the incoming light" should be irrelevant.

L203: Why is the uncertainty related to the recorded signal I0 negligible? When approaching the detection limit, I0 is affected by noise, so the uncertainty goes to infinity as the detection limit is approached.

L206: Eq. (1) does no look correct. It should be I1 = (I0 - D) / A. The dark current should be measured at the same amplification as the "light" current. So if there is no light entering the PMT, I0 and D should be identical. Since the dark current is not zero (e.g., because of Schott noise) and scales with the amplification, I1 can only be zero when no radiation is falling on the diffuser if I0 and D are divided by the same factor A. If the authors use indeed their Eq. (1) to calculate I1, the error would be small if I0 is much higher than D. However, there would be a significant error at very low light levels when I0 is only somewhat larger than D.

Line 224: y is presumably the correction factor cf, and x is presumably temperature theta in °C. If so, the equitation should be written: cf = a * theta + b. Also, I find it awkward to use a symbol with two letters to describe a factor. Why not use "c" instead of "cf"?

Line 226: If theta is the symbol for temperature, it should also be used in Eq. (2).

L230: This is only justified if the PTFE diffuser material used by Y&S is the same as that used by the authors. Is this the case? What is the evidence that both diffuser materials have the same temperature dependence?

L231: "Since the 125th Day of the Year (DOY), in May of 2017". Just say: "Since 5 May 2017 ..."

L255 and L258: I am surprised that distances as close as 5 or 7 cm between lamp and

diffuser are considered. At 5 cm, a 1 mm uncertainty in the lamp-diffuser distance results in an uncertainty of about a 4% in the irradiance at the diffuser. Since the authors like to detect differences of less than 0.5%, the lamp-diffuser distance would have to be reproducible by 0.12 mm. Is the coupling between the lamp housing and the diffuser really that reproducible? The author should comment on the trade-off between short distance (= low noise) and long distance (= less uncertainty from distance inaccuracies and less uncertainty from heating the diffuser).

Figure 3 and lines 280, 281, 287 Figure 3 does not show a "correction factor". Instead it shows the cosine error of the diffuser, expressed in measured angular response divided by the ideal (=cosine) response, for illumination by a point source and for isotropic illumination. The term "correction factor" is highly confusing because it implies that the values shown in Figure 3 are the values by which solar measurements have to be scaled to correct for the diffuser's cosine error. However, such a correction would also depend on wavelength, as correctly discussed in Section 3.3.2.

L282: "distributed uniformly in the horizon" > "isotropic"

Figure 4c: I don't understand why the error increases sharply between 85° and 88° considering that solar radiation becomes more diffuser over this angular range and the lack of a large spike in the angular response at these angles, according to Figure 3. It seems to me that the spike shown in Figure 4c is a result of interpolation artifacts of the cosine error due to the fact that the cosine error at 90° is infinite.

Section 3.5. The experiments described in this section do not characterize non-linearity. Non-linearity means that the output of an instrument (in this case the signal I2) varies linearly with the input (i.e., spectral irradiance). An experimental setup for testing non-linearity could, for example, involve a light source whose intensity can be changed over a wide range (ideally the same range as covered by solar radiation) plus a beam splitter that would direct half of the light towards the Bentham and the other half on a second radiometer whose linearity has been confirmed by other means. By comparing

the signal of the Bentham with that of the secondary (linear) radiometer, departures from linearity can be determined. (There are also other means to characterize a radiometer for non-linearity, but the test above illustrates the principle sufficiently.) PMTs in photon-counting mode tend to be non-linear if two photos arrive within the dead-time of the counter, but this is not the case here.

Instead, the experiments described in Section 3.5 tested something different. They determined whether the responsivity of the system changes (either permanently or temporarily) if the system's PMT is temporarily exposed to either high light levels, or is producing large currents due to a unreasonably high high-voltage setting. The term "hysteresis" comes to mind describing the effect but it not the ideal term either. Perhaps using descriptive headings would be the best. For example, "Section 3.5, Non-linearity" could be replaced with "Section 3.5. Change in responsivity after exposure to high radiation levels". Likewise, Subsection 3.5.1 and 3.5.2 could be replaced with "Change in responsivity during usual operating conditions" and "Change in responsivity after exposure to unusually high radiation levels", respectively. The word "non-linearity" in the text of Section 3.5. should also be replaced with a more descriptive term such as "change in responsivity".

L367: The maximum photocurrent of a PMT of the type used by the authors is typically 1000 nA. It should be noted that the PMT was operated well beyond the recommended range.

L368, sentence starting with "Two possible . . .": What does this sentence refer to? The period when photocurrent was > 15000, or the period after adjusting the HV to 400 V resulting in photocurrents < 500 nA? Also the phrase "of similar instrument" is strange. It suggests that problems resulting from PMT overexposure are about to be described in general terms. Instead, the rest of paragraph addresses only problems observed with the Bentham.

Line 370: I don't understand "the day following the intensity of the recorded signal".

L395: I don't understand "However, in this case the response of the AAO gradually increased while we were measuring the lamps irradiance at noon." AAO standards for "Aosta-Saint Christophe". Do you mean the responsivity of the system located at AAO increased? Also "gradually" implies that you measured the lamp several times, and there was a clear trend in responsivity. Was this the case?

L407: The maximum photocurrent during these experiments was apparently 2500 nA, resulting in a 3% effect. However, in line 368 it is noted that photocurrents as high 20000 nA were measured before July 2006. This suggests that measurements during this period are likely affected by saturation. Is this the reason why the dataset (e.g. Fig. 9) does not include measurements prior to July 2006?

L440: Explain acronym "KS"

Section 4.1: Explain that spectral measurements are traceable to the scale of spectral irradiance PTB. I assume that the PTB has realized their scale more than once. The specific PTB scale used for the measurements of the network should be referenced.

L466: "in units of electrical intensity (nA)" > "in units of nA"

L482: Why 320 nm? 315-400 nm is now the standard range for UV-A.

Line 490 references Mayer and Kylling, 2005 for the UVSPEC model while line 304 references Emde et al., 2016. Were two different model versions used? If not, perhaps both papers could be cited on line 304 and the reference in line 490 could be omitted.

L523: "corresponds to the minimum response of the diffuser " The diffuser does not have a responsivity. What's likely meant is that the responsivity of the system is smaller when the diffuser's temperature is lower. Please clarify.

L535: "even the Level 1.5 spectra are of good quality." The sentence suggests that Level 1.5 data are less accurate than Level 1.0 data, which is likely not the case. Rephrase or delete "even" at the least.

L554: "by the company" What company? ; "quality certificate." > "calibration certificate."

L558 - L593: The text in these paragraphs is very hard to follow. There are changes in the calibration, which were later reversed when new information became available. I guess this information is of little relevance for the average reader and only of importance to the authors for documenting the adjustments in calibration factors. Hence, there may be no need to improve the text. Still, I am wondering whether the information could be presented in an easier-to-digest format. Perhaps a table could be added that describes the reasons for the differences in Level 2 and Level 1 data for the different periods discussed in the text.

Caption Figure 8: The sentence "Shaded area represents $\pm2\%$ from perfect agreement between Bentham5541 and QASUME" is likely not correct. The $\pm2\%$ range is indicate by two lines. The shaded areal is likely "the combined expanded uncertainty of the Bentham5541 Level 2 and the QASUME spectra for each intercomparison" as indicated in line 613.

L630: Add "for most years" after "There is a clear improvement of the results when the Level 2 dataset is used." The difference in the results of both instruments is worse for Level 2 data in 2006, 2011, and 2013.

Section 5.1.5.: Non-linearity was not tested. The sentence should be changed to: "The responsivity of the Bentham5541 does not change by more than 0.5% if the system is exposed for short times to high radiation levels that may occur during clear skies in summer months. The resulting uncertainty is set to zero because it is part of the uncertainty from instrument instability (Sect. 5.1.3)."

L750: I recall from reading Section 3.5 that the responsivity my change by up to 0.5% after overexposure. So the uncertainty cannot not be 0%. If the uncertainty is part of another uncertainty component, this should be stated.

Caption Table 2: According to the caption, numbers are "standard uncertainties". I suspect that this is incorrect and that expanded uncertainties for k=2 are shown. But I may be wrong. In either case, since Table 2 presents the final uncertainty budget, expanded uncertainties should be shown. All number should also be larger than the expanded uncertainties shown in Table 1 because of additional uncertainty components affecting solar measurements.

L773 - L778: Also uncertainties in this section should be "expanded" uncertainties.

L790: "is linear, even for very high signal levels." No. There was no real test of non-linearity. Only changes in responsivity following overexposure were tested - see my comments above. Having said this, I believe that the system is indeed linear. Otherwise, the comparison between the Bentham and QASUME would have indicated differences as a function of solar irradiance. This could be mentioned.

L796 - 800: It is not clear whether the uncertainties mentioned here are expanded uncertainties or not.

L861 and L52: Change year from 2013 to 1994.

Specific comments to data available at https://doi.org/10.5281/zenodo.3934324:

The paper includes the link https://doi.org/10.5281/zenodo.3934324 that points to the actual data. The associated website on ZENODO is well organized, however, a few shortcomings and errors should be fixed:

- Data files are compressed in zip files. The naming convention of these zip files should be added to the website.

- The readme.txt file that describes the data files includes the sentence: "*Spectral scans begin at 290 nm and usually end at 400 nm. However, scans may also end at 400 nm." I presume "may also end at 400 nm." should be "may also end at 500 nm."

- The same readme file also includes the line: "Columns 6 - 446: Irradiance (in Watt/m^2/nm) at wavelengths 290 - 500 nm respectively with a step of 0.25 nm" The

number of wavelengths between 290 and 500 in 0.25 nm steps should be 841, not 446.

Language:

Language - General:

(Important!) Change "response" (i.e. the term describing the ratio of signal and irradiance) to "responsivity" throughout the paper. See for example: https://en.wikipedia.org/wiki/Responsivity

"dependence from" > "dependence on"

"In a different day" > "On a different day"

"of the order of" > "in the order of"

"data was" > "data were" (data is plural)

"which mediates" > "which lies" or just delete, or reword sentence.

"Huelsen" > "Hülsen"

Always include a space between number and unit (e.g., 570m > 570 m)

"statistic noise" > "statistical noise"

Language - Specific:

L10: "in the North-western" > "in the North-Western region"

L14: "moreover" > "also"

L18: "The used Quality" > "The Quality"

L21: "consist one of" > "consist of one of"

L37: "exposed to more or to less" > "exposed to either more or to less"

L40: Delete "and on time"

L46: "the North and the South hemisphere in spring in the 1980s," > "both hemispheres during spring since the 1980s"

L48: "also experienced" > "were also observed"

L69: Delete "However,"

L83: "certifies" > "confirms" (To certify something, you typically have to be accredited by a licensing board or a standards organization.)

L96: tall > high

L128: "consists from" > "consists of"

L149: "driven" > "coupled" or "guided"

L305: Angstrom > Ångström

Section 3.4: The word "head" for describing the metal piece at the end of the fiber is awkward. I suggest "termination". Line 333: Change "on the first head and exits the fiber through a second aperture on the second head." to "at one end and exits the fiber through a second aperture on the other end."

L342: The sentence "in order to detect possible azimuthal dependence of the response, which would show that there is a problem with the azimuth response," is a tautology. Just delete the second part of the phrase.

L345: "bellow" > "below"

L349, 353: "bubble" > "bubble level"

L354, 689: "After begin" > "After" L377: "at early morning," > "in the early morning"

L442: "basis at the facilities of the PMOD- WRC" > "basis to PMOD-WRC"

L448: "the three working standard 200 Watt lamps" > "the three 200 Watt working standards"

L467: Rephrase "divided with the used level of amplification."

L479, 504: "DOY 125 of 2017" > "5 May 2017"

L539: "characterized" > "classified"

L549: "however exceed" > "exceed"

L667: "standard throughout the years" > "identical for all years"

L680: "to the accredited" > "at accredited"

L697: "Sect. 3.5 resulted that" > "Sect. 3.5 suggests that"

L725: delete "and certify"

L733: "Thus, a strict calculation of the uncertainty would demand to take" > "Thus, a rigorous calculation of the uncertainty would require to take"

Caption Table 1: "(in %) o the Level 2" > "(in %) for Level 2"

L782: "range 290 – 500 nm" > "range of 290 – 500 nm"

L789: "proved through" > "confirmed by"

L790: "It has been also proved that" > "It has also been shown that"

L795: "have significant contribution to" > "contribute significantly to"

L812: "when at the UV index for at least 15 days of the month is available." > "when the noon-time UV index is available for at least 15 days per month."

---

## Referee Comment (RC2) · Anonymous Referee #2 · 3 Sep 2020

General comments: The manuscript describes procedures and characteristics related to quality control (QC) and quality assurance (QA) of the reference spectroradiometer of the Italian UV network. The uncertainty budget related to radiometric uncertainty and the overall uncertainty of UV measurements is presented. As a result, a unique homogenized dataset is made available. Different data levels have been explained as well as reasons for differences between them. The performance of the instrument has been monitored by regular comparison campaign with the QASUME reference spectroradiometer from PMOD-WRC. The manuscript includes all relevant uncertainty sources and they are adequately presented. The importance of the work and the reference spectroradiometer measurements are well described and can serve as example for other national UV networks. It would have been good to see the transfer of the cali-

bration to the other instruments of the network and discussion of uncertainties related to the transfer process and the effect to the overall uncertainty of the site measurements. I am not a native English speaker, but I think the language can be improved. I think the manuscript is important for the UV scientific community and represent state of the art of QC/QA of solar UV measurements. The long UV time series is of high quality and multidisciplinary communities can benefit from it. I recommend the article to be published in Earth System Science Data.

Specific comments:

l. 34-38: Is it really so that people are adapted to proper sun-exposure behaviour? Do you have an example? I think there exist still many problems at areas of high exposure, e.g. cataracts in Tibet.

l. 46: Please check if Solomon et al. 1986 describes the Northern ozone depletion. If not, please find a reference for the Arctic ozone depletion and check the decade of the first signs of arctic spring-time ozone depletion.

l. 69: "for SZAs below 75° and wavelengths below 305 nm." Do you mean ABOVE 305 nm?

l. 100: "extremely high levels of 100the UV irradiance" , please add what are the extreme values in UV index (some number).

l. 132: within less THAN 0.5 °C?

l. 134: please discuss whether the humidity difference (10% in winter, 60% in summer) can introduce errors/uncertainties in measurements.

l. 164: at Saint Christophe -> at AAO?

l. 171: Do you have a web page for UV index prediction? If yes, please show the web address.

l. 186: If the Instrument characterization and measurement corrections of Section 3

are only about the reference spectroradiometer, please indicate it in the header of the Section 3

l. 194: Please add a sentence or two explaining what is dark current and from where does it come from.

l. 208: Teflon diffuser: In line 146 you write that the instrument has a quartz diffuser?

l. 224: Eq.3. cf=

l. 236-238: It is unclear what you actually did and how did you end up to the result. Please rephrase.

Section 3.2.: Please explain what is the typical effect (in %) of the temperature correction of the Teflon diffuser for a dataset during a winter month and a summer month.

l. 278: For which wavelengths are the results showed? Any wavelength dependency in the angular response?

l. 300: Header: Modelling the errors due to what, angular response? Please change the header to be more explanatory.

l 309: "Simulations for Davos were performed in order to show that at such high altitudes the error becomes more important", please explain why this is the case.

Section 3.5.1., l.370. I don't really understand how you can address this point by 200W lamp measurements as your 200W lamp has always the same intensity/irradiance, hasn't it? Shouldn't you vary the intensity of the lamp in order to address the point 2?

The same applies for Section 3.5.2., or do you mean that your PMT has a "memory"?

Section 4.1.: How do you take into account the possible drift of your 200W lamps? Do you rotate them, as you have three lamps?

Section 4.2.4: It is a little bit diffucult to follow what has happened during each p-period. Anyway, the message is unclear: is the Level 2 time series homogeneous? If

yes, which calibration scale has been used? Have you used a step-wise calibration change in calibration: change after each calibration? Have you taken into account that the instrument's reponse may have changed slowly between two calibration, and have this been taken into account e.g. by linear interpolation between two calibrations?

Fig 8. Please check the Figure Caption.

l. 630, This is not true for all years (2006 and 2013).

Section 5.2.5. I miss a table with all the uncertainty sources contributing the overall uncertainty (like you have in Table 1 for radiometric uncertainty).

---

## Author Comment (AC1) · 29 Sep 2020

We would like to acknowledge anonymous reviewer#1 for her/his constructive comments which helped us improve the manuscript substantially. Our responses follow the reviewer's comments (in bold). Since page and line numbers of the original manuscript are different in the new version, the new page and line numbers (in the version with marked changes) are also given where needed.

**Specific comments**

**L22: "expanded uncertainty": provide the coverage factor (likely k = 2)**

**Reply**

Done

**Figure 1: It would be much better to show a topographical map instead of a political map.**

**Reply**

The political map has been replaced by a topographical map.

**L174: "As already discussed the Bentham5541 is traceable to world reference QASUME." I would say the measurements are traceable to the scale of spectral irradiance established by PTB. At least the chain shown in Figure 5 starts with PTB. QASUME is only used for QA/QC, but data are not scaled to match QASUME's measurements. The role of QASUME should be better described.**

**Reply**

Paragraph 2.4 has been changed according to the reviewer suggestions, and it is clear that measurements are traceable to the spectral irradiance scale established by PTB.

**Section 3.1: either use the term "dark current" or "dark signal" but not a mixture of both.**

**Reply**

Only the term "dark signal" is used throughout the manuscript in the revised version.

**L195: Why would the dark current depend on "the level of the intensity of the incoming light"? The dark current is presumably measured with a physical shutter closed, or perhaps determined by scanning below 280 nm where no solar radiation can be detected at the Earth's surface (presuming that there is no stray light). So the "level of the intensity of the incoming light" should be irrelevant.**

**Reply**

The reviewer is right. The phrase "the level of the intensity of the incoming light" has been removed (line 224).

**L203: Why is the uncertainty related to the recorded signal I0 negligible? When approaching the detection limit, I0 is affected by noise, so the uncertainty goes to infinity as the detection limit is approached.**

**Reply**

The reviewer is again right. However, already from the originally submitted version of the manuscript we have clearly specified that we refer to conditions under which the measured signal is at least two orders of magnitude higher relative to the dark signal (i.e. SZA<85 degrees and wavelengths above 305 nm) (lines 225 - 227). Nevertheless a sentence clarifying that the uncertainties in the dark signal become more important as the detection limit is approached has been added (lines 227 - 229).

**L206: Eq. (1) does not look correct. It should be I1 = (I0 - D) / A. The dark current should be measured at the same amplification as the "light" current. So if there is no light entering the PMT, I0 and D should be identical. Since the dark current is not zero (e.g., because of Schott noise) and scales with the amplification, I1 can only be zero when no radiation is falling on the diffuser if I0 and D are divided by the same factor A. If the authors use indeed their Eq. (1) to calculate I1, the error would be small if I0 is much higher than D. However, there would be a significant error at very low light levels when I0 is only somewhat larger than D.**

**Reply**

For the measurement of the dark signal we use Equation 1 as it is in the original version of the manuscript. The dark signal is measured without using amplification (i.e. amplification=1). So, the equation is correct. Even if the fact that we use a different amplification level for the measurement of the dark signal and the "light" signal, introduces additional uncertainties when the "light" signal levels are very low (i.e. at very short wavelengths, below 300 nm, when SZAs is above 75°), the uncertainties at such low signal levels are not important for usual applications of the data (e.g. climatological analysis, calculation of daily doses, information of the public).

**Line 224: y is presumably the correction factor cf, and x is presumably temperature theta in ∘C. If so, the equitation should be written: cf = a * theta + b. Also, I find it awkward to use a symbol with two letters to describe a factor. Why not use "c" instead of "cf"?**

**Reply**

The proposed changes have been applied to Equation 3.

**Line 226: If theta is the symbol for temperature, it should also be used in Eq. (2).**

**Reply**

Done.

**L230: This is only justified if the PTFE diffuser material used by Y&S is the same as that used by the authors. Is this the case? What is the evidence that both diffuser materials have the same temperature dependence?**

**Reply**

Indeed the diffuser used by Bentham 5541 is a Teflon diffuser of similar thickness as one of the diffusers tested by Ylianttila and Schreder (2005). Relative information has been added at the beginning of Sect. 3.2.

**L231: "Since the 125th Day of the Year (DOY), in May of 2017". Just say: "Since 5 May 2017 . . ."**

**Reply**

The proposed correction has been applied to the manuscript.

**L255 and L258: I am surprised that distances as close as 5 or 7 cm between lamp and diffuser are considered. At 5 cm, a 1 mm uncertainty in the lamp-diffuser distance results in an uncertainty of about a 4% in the irradiance at the diffuser. Since the authors like to detect differences of less than 0.5%, the lamp-diffuser distance would have to be reproducible by 0.12 mm. Is the coupling between the lamp housing and the diffuser really that reproducible? The author should comment on the trade-off between short distance (= low noise) and long distance (= less uncertainty from distance inaccuracies and less uncertainty from heating the diffuser).**

**Reply**

The distances of 7 and 20 cm (for the short and the long setup of the calibrator) which were reported at the original version of the manuscript were wrong. The corresponding (correct) distances reported in the new version of the manuscript are 12 and 30 cm respectively. However, the reproducibility of the distance (commented by the reviewer for the distance of 7 cm) is still very significant even for the distance of 12 cm. A new section (Sect. 5.1.10) where the relative uncertainties are discussed has been added to the manuscript. These uncertainties have been also added to the overall uncertainty budget (Table 1). The overall uncertainty budget has not changed since the additional uncertainty is small. Discussion regarding the uncertainty due to different noise levels and heating of the diffuser already existed (Sections 5.1.4 and 5.1.8 in the original version of the manuscript).

**Figure 3 and lines 280, 281, 287 Figure 3 does not show a "correction factor". Instead it shows the cosine error of the diffuser, expressed in measured angular response divided by the ideal (=cosine) response, for illumination by a point source and for isotropic illumination. The term "correction factor" is highly confusing because it implies that the values shown in Figure 3 are the values by which solar measurements have to be scaled to correct for the diffuser's cosine error. However, such a correction would also depend on wavelength, as correctly discussed in Section 3.3.2.**

**Reply**

The discussion in Section 3.3.1 and the legend of figure 3 have been updated following the recommendations of the reviewer. Furthermore, the discussion for the dependence of the cosine error from wavelength has been updated and moved earlier to the same paragraph (lines 367 - 371) in order to assist the relative discussion.

**L282: "distributed uniformly in the horizon" > "isotropic"**

**Reply**

Done

**Figure 4c: I don't understand why the error increases sharply between 85∘ and 88∘ considering that solar radiation becomes more diffused over this angular range and the lack of a large spike in the angular response at these angles, according to Figure 3. It seems to me that the spike shown in Figure 4c is a result of interpolation artifacts of the cosine error due to the fact that the cosine error at 90∘ is infinite.**

**Reply**

At 88∘ the cosine error (for illumination by a point source) is 38% and is outside the limits of the y-axis (in Figure 3). We kept the y-axis between 0.96 and 1.08 in order to give a better picture of the cosine errors at lower SZA since above 85° the cosine error is not significant (at least for the measurements performed at Aosta). This information (about the error of 38% at 88° SZA) has been added in Section 3.3.1. According to the simulations, at Davos, the direct component at 495 nm is ~12% of the total solar irradiance measured for SZA=88°. These numbers justify the error of 5% in the measured irradiance shown in Figure 4c.

**Section 3.5. The experiments described in this section do not characterize nonlinearity. Non-linearity means that the output of an instrument (in this case the signal I2) varies linearly with the input (i.e., spectral irradiance). An experimental setup for testing non-linearity could, for example, involve a light source whose intensity can be changed over a wide range (ideally the same range as covered by solar radiation) plus a beam splitter that would direct half of the light towards the Bentham and the other half on a second radiometer whose linearity has been confirmed by other**

means. By comparing the signal of the Bentham with that of the secondary (linear) radiometer, departures from linearity can be determined. (There are also other means to characterize a radiometer for non-linearity, but the test above illustrates the principle sufficiently.) PMTs in photon-counting mode tend to be non-linear if two photons arrive within the dead-time of the counter, but this is not the case here.

Instead, the experiments described in Section 3.5 tested something different. They determined whether the responsivity of the system changes (either permanently or temporarily) if the system's PMT is temporarily exposed to either high light levels, or is producing large currents due to a unreasonably high high-voltage setting. The term "hysteresis" comes to mind describing the effect but it not the ideal term either. Perhaps using descriptive headings would be the best. For example, "Section 3.5, Non-linearity" could be replaced with "Section 3.5. Change in responsivity after exposure to high radiation levels". Likewise, Subsection 3.5.1 and 3.5.2 could be replaced with "Change in responsivity during usual operating conditions" and "Change in responsivity after exposure to unusually high radiation levels", respectively. The word "non-linearity" in the text of Section 3.5. should also be replaced with a more descriptive term such as "change in responsivity"

**Reply**

The reviewer is right again. Indeed "Change in responsivity after exposure to high radiation levels" is more accurate for what we have investigated here instead of "non-linearity". The whole section has been modified following the suggestions of the reviewer. A small sub-section (3.5.3) has also been added where the non-linearity issue (as also defined by the reviewer in his comments) is discussed.

**L367: The maximum photocurrent of a PMT of the type used by the authors is typically 1000 nA. It should be noted that the PMT was operated well beyond the recommended range.**

**Reply**

Relative information has been added in lines 458 – 460

**L368, sentence starting with "Two possible . . .": What does this sentence refer to? The period when photocurrent was > 15000, or the period after adjusting the HV to 400 V resulting in photocurrents < 500 nA? Also the phrase "of similar instrument" is strange. It suggests that problems resulting from PMT overexposure are about to be described in general terms. Instead, the rest of paragraph addresses only problems observed with the Bentham**

**Reply**

The particular sentence (now in line 469) has been changed to:

"Two problems which have affected the Bentham5541 spectroradiometer during its regular operation before July 2006 are the following:"

So now the meaning of the sentence is clearer.

**Line 370: I don't understand "the day following the intensity of the recorded signal".**

**Reply**

1. The sentence has been changed and additional information was added (lines 473 - 476): **"**The responsivity was changing during the day following the intensity of the recorded signal. In this case the responsivity of the PMT decreased after exposure to very high signal. Then it gradually increased again until the next spectral scan begun. The responsivity in this case changes during the day depending on the resting time between consecutive scans, and the maximum intensity of the recorded signal during each scan."

**L395: I don't understand "However, in this case the response of the AAO gradually increased while we were measuring the lamps irradiance at noon." AAO standards for "Aosta-Saint Christophe". Do you mean the responsivity of the system located at AAO increased? Also "gradually" implies that you measured the lamp several times, and there was a clear trend in responsivity. Was this the case?**

**Reply**

It was a typo. "AAO" has been replaced with "Bentham5541". The word "gradually" was inaccurate and confusing and has been deleted (lines 504 - 505)

**L407: The maximum photocurrent during these experiments was apparently 2500 nA, resulting in a 3% effect. However, in line 368 it is noted that photocurrents as high 20000 nA were measured before July 2006. This suggests that measurements during this period are likely affected by saturation. Is this the reason why the dataset (e.g. Fig. 9) does not include measurements prior to July 2006?**

**Reply**

The reviewer is right. Relative information has been also added in the manuscript (lines 524 - 526).

**L440: Explain acronym "KS"**

**Reply**

Explanation was added in line 559.

**Section 4.1: Explain that spectral measurements are traceable to the scale of spectral irradiance PTB. I assume that the PTB has realized their scale more than once. The specific PTB scale used for the measurements of the network should be referenced.**

**Reply**

The relative information has been added to the manuscript (lines 564 - 566)

**L466: "in units of electrical intensity (nA)" > "in units of nA"**

**Reply**

Done

**L482: Why 320 nm? 315-400 nm is now the standard range for UV-A.**

**Reply**

It was a typo. The range is 315 – 400 nm (line 611)

**Line 490 references Mayer and Kylling, 2005 for the UVSPEC model while line 304 references Emde et al., 2016. Were two different model versions used? If not, perhaps both papers could be cited on line 304 and the reference in line 490 could be omitted.**

**Reply**

Emde et al., 2016 is now used as reference throughout the manuscript.

**L523: "corresponds to the minimum response of the diffuser " The diffuser does not have a responsivity. What's likely meant is that the responsivity of the system is smaller when the diffuser's temperature is lower. Please clarify**

**Reply**

When temperature of the diffuser changes, its transmissivity also changes (as suggested by Ylianttila and Schreder ,2004). Indeed "response" was not the most appropriate word here. Now the word "transmissivity" is used.

**L535: "even the Level 1.5 spectra are of good quality." The sentence suggests that Level 1.5 data are less accurate than Level 1.0 data, which is likely not the case. Rephrase or delete "even" at the least.**

**Reply**

The word "even" has been deleted (line 665)

**L554: "by the company" What company? ; "quality certificate." > "calibration certificate."**

**Reply**

The name of the company which provided the lamps (Schreder CMS) has been added to the manuscript (line 690). The suggested correction has been also applied.

**L558 - L593: The text in these paragraphs is very hard to follow. There are changes in the calibration, which were later reversed when new information became available. I guess this information is of little relevance for the average reader and only of importance to the authors for documenting the adjustments in calibration factors. Hence, there may be no need to improve the text. Still, I am wondering whether the information could be presented in an easier-to-digest format. Perhaps a table could be added that describes the reasons for the differences in Level 2 and Level 1 data for the different periods discussed in the text**

**Reply**

The particular part of the manuscript (Section 4.2.4) has been shortened and simplified. Information which was not of interest for most readers has been removed. A small paragraph summarizing the differences between Level 1 and Level 2 spectra has been added at the end of the section.

**Caption Figure 8: The sentence "Shaded area represents ±2% from perfect agreement between Bentham5541 and QASUME" is likely not correct. The ±2% range is indicated by two lines. The shaded areal is likely "the combined expanded uncertainty of the Bentham5541 Level 2 and the QASUME spectra for each intercomparison" as indicated in line 613.**

**Reply**

The caption has been corrected

**L630: Add "for most years" after "There is a clear improvement of the results when the Level 2 dataset is used." The difference in the results of both instruments is worse for Level 2 data in 2006, 2011, and 2013.**

**Reply**

Done (line 784)

**Section 5.1.5.: Non-linearity was not tested. The sentence should be changed to: "The responsivity of the Bentham5541 does not change by more than 0.5% if the system is exposed for short times to high radiation levels that may occur during clear skies in summer months. The resulting uncertainty is set to zero because it is part of the uncertainty from instrument instability (Sect. 5.1.3)."**

**Reply**

The sentence has changed to:" There is no detectable change in the responsivity of Bentham5541 when it is exposed to high radiation levels that may occur during clear skies in summer months. There is also no sign of non linearity in the measurements (as already explained in Sect. 3.5). Nevertheless, even if there is some uncertainty related to these phenomena it is set to zero because it is part of the uncertainty from instrument instability (Sect. 5.1.3)." We slightly modified the sentence suggested by the reviewer for the reasons explained in the answer to the next comment.

**L750: I recall from reading Section 3.5 that the responsivity my change by up to 0.5% after overexposure. So the uncertainty cannot not be 0%. If the uncertainty is part of another uncertainty component, this should be stated.**

**Reply**

This comment is not accurate. In section 3.5 we show that the responsivity changes up to 0.5% when the signal is 3 – 4 times higher than the maximum levels recorded at Aosta. For usual operational conditions no change in the responsivity was found. Nevertheless, even if there is any issue of non- linearity or change in the responsivity that we were not able to detect, it is implicitly taken onto account in the uncertainty related to the instrument instability (as the reviewer suggested and as it is now stated in line 861).

**Caption Table 2: According to the caption, numbers are "standard uncertainties". I suspect that this is incorrect and that expanded uncertainties for k=2 are shown. But I may be wrong. In either case, since Table 2 presents the final uncertainty budget, expanded uncertainties should be shown.**

**All number should also be larger than the expanded uncertainties shown in Table 1 because of additional uncertainty components affecting solar measurements.**

**Reply**

They were standard uncertainties. Following the reviewer suggestion they have been changed to expanded uncertainties.

**L773 - L778: Also uncertainties in this section should be "expanded" uncertainties.**

**Reply**

Expanded uncertainties are provided instead of standard uncertainties in the revised version of the manuscript as the reviewer suggested (lines 955 – 956 and 962 - 965).

**L790: "is linear, even for very high signal levels." No. There was no real test of nonlinearity. Only changes in responsivity following overexposure were tested - see my comments above. Having said this, I believe that the system is indeed linear. Otherwise, the comparison between the Bentham and QASUME would have indicated differences as a function of solar irradiance. This could be mentioned.**

**Reply**

The particular phrase has been changed to **"**It has also been shown that the responsivity of the instrument does not change, even after exposure to very high signal levels" In the revised version (line 977). Furthermore, we added one more paragraph at Section 3.5 adding the information recommended by the reviewer:

 "3.5.3 Linearity

Comparison of the spectral measurements from Bentham5541 with simultaneous measurements of QASUME during recent inter-comparison campaigns (2015, 2017, 2019), as well as with measurements from broad-band instruments operating at AAO did not yield any sign of detectable no linearity of the Bentham5541. Thus, even if there is any non-linearity effect the relative uncertainty is very small relative to the overall uncertainties in the measurements. "

**L796 - 800: It is not clear whether the uncertainties mentioned here are expanded uncertainties or not.**

**Reply**

They are expanded uncertainties. We added the necessary information.

**L861 and L52: Change year from 2013 to 1994.**

**Reply**

Done

**Specific comments to data available at https://doi.org/10.5281/zenodo.3934324:**

**The paper includes the link https://doi.org/10.5281/zenodo.3934324 that points to the actual data. The associated website on ZENODO is well organized, however, a few shortcomings and errors should be fixed:**

**- Data files are compressed in zip files. The naming convention of these zip files should be added to the website.**

**- The readme.txt file that describes the data files includes the sentence: "*Spectral scans begin at 290 nm and usually end at 400 nm. However, scans may also end at 400 nm." I presume "may also end at 400 nm." should be "may also end at 500 nm."**

**- The same readme file also includes the line: "Columns 6 - 446: Irradiance (in Watt/m^2/nm) at wavelengths 290 - 500 nm respectively with a step of 0.25 nm" The number of wavelengths between 290 and 500 in 0.25 nm steps should be 841, not 446**

**Reply**

All suggested changes have been applied to the readme.txt file. A new version of the dataset (the only change relative to the previous one is the txt file which has been updated) is available at https://doi.org/10.5281/zenodo.4028907 . The manuscript and the reference list have been updated properly.

**Language - General:**

**(Important!) Change "response" (i.e. the term describing the ratio of signal and irradiance) to "responsivity" throughout the paper. See for example: https://en.wikipedia.org/wiki/Responsivity**

**"dependence from" > "dependence on" "**

**In a different day" > "On a different day"**

**"of the order of" > "in the order of"**

**"data was" > "data were" (data is plural)**

**"which mediates" > "which lies" or just delete, or reword sentence.**

**"Huelsen" > "Hülsen"**

**Always include a space between number and unit (e.g., 570m > 570 m)**

**"statistic noise" > "statistical noise"**

**Reply**

All the suggested Language – general corrections have been incorporated in the manuscript.

**Language - Specific:**

**L10: "in the North-western" > "in the North-Western region"**

**L14: "moreover" > "also"**

**L18: "The used Quality" > "The Quality"**

**L21: "consist one of" > "consist of one of"**

**37: "exposed to more or to less" > "exposed to either more or to less"**

**L40: Delete "and on time**

**L46: "the North and the South hemisphere in spring in the 1980s," > "both hemispheres during spring since the 1980s"**

**L48: "also experienced" > "were also observed"**

**L69: Delete "However,"**

**L83: "certifies" > "confirms" (To certify something, you typically have to be accredited by a licensing board or a standards organization.)**

**L96: tall > high**

**L128: "consists from" > "consists of"**

**L149: "driven" > "coupled" or "guided"**

**L305: Angstrom > Ångström**

**Section 3.4: The word "head" for describing the metal piece at the end of the fiber is awkward. I suggest "termination".**

**Line 333: Change "on the first head and exits the fiber through a second aperture on the second head." to "at one end and exits the fiber through a second aperture on the other end."**

**L342:** The sentence "in order to detect possible azimuthal dependence of the response, which would show that there is a problem with the azimuth response," is a tautology. Just delete the second part of the phrase.

**L345:** "bellow" > "below"

**L349, 353:** "bubble" > "bubble level"

**L354, 689:** "After begin" > "After"

**L377:** "at early morning," > "in the early morning"

**L442:** "basis at the facilities of the PMOD- WRC" > "basis to PMOD-WRC"

**L448:** "the three working standard 200 Watt lamps" > "the three 200 Watt working standards"

**L467:** Rephrase "divided with the used level of amplification."

**L479, 504:** "DOY 125 of 2017" > "5 May 2017"

**L539:** "characterized" > "classified"

**L549:** "however exceed" > "exceed"

**L667:** "standard throughout the years" > "identical for all years"

**L680:** "to the accredited" > "at accredited"

**L697:** "Sect. 3.5 resulted that" > "Sect. 3.5 suggests that"

**L725:** delete "and certify"

**L733:** "Thus, a strict calculation of the uncertainty would demand to take" > "Thus, a rigorous calculation of the uncertainty would require to take" Caption Table 1: "(in %) o the Level 2" > "(in %) for Level 2"

**L782:** "range 290 – 500 nm" > "range of 290 – 500 nm"

**L789:** "proved through" > "confirmed by"

**L790:** "It has been also proved that" > "It has also been shown that"

**L795:** "have significant contribution to" > "contribute significantly to"

[revised manuscript text omitted]

---

## Author Comment (AC2) · 29 Sep 2020

We would like to thank anonymous reviewer#2 for her/his constructive comments which helped us improve the manuscript. Our responses follow the reviewer's comments (in bold). Since page and line numbers of the original manuscript are different in the new version, the new page and line numbers (in the version with marked changes) are also given where needed.

**General comments:**

**The manuscript describes procedures and characteristics related to quality control (QC) and quality assurance (QA) of the reference spectroradiometer of the Italian UV network. The uncertainty budget related to radiometric uncertainty and the overall uncertainty of UV measurements is presented. As a result, a unique homogenized dataset is made available. Different data levels have been explained as well as reasons for differences between them. The performance of the instrument has been monitored by regular comparison campaign with the QASUME reference spectroradiometer from PMOD-WRC. The manuscript includes all relevant uncertainty sources and they are adequately presented. The importance of the work and the reference spectroradiometer measurements are well described and can serve as example for other national UV networks. It would have been good to see the transfer of the calibration to the other instruments of the network and discussion of uncertainties related to the transfer process and the effect to the overall uncertainty of the site measurements. I am not a native English speaker, but I think the language can be improved. I think the manuscript is important for the UV scientific community and represent state of the art of QC/QA of solar UV measurements. The long UV time series is of high quality and multidisciplinary communities can benefit from it. I recommend the article to be published in Earth System Science Data.**

**Reply**

The present document is focused to the description of the processes which ensure the high quality of the solar spectra measured at Aosta-Saint Christophe by the Bentham spectroradiometer, which is the reference instrument for the whole regional network of the Aosta Valley (which s a regional, and not a national network). Analytical discussion regarding the quality of the measurements of all instruments of the network is out of the scope of the present document.

Furthermore, the re-evaluation of the calibration factors of all instruments of the network, and the quantification of the relative uncertainties is still an ongoing process which will be completed in the following months. Thus, we are not able at this moment to provide accurate information for the quality of the re-evaluated data for the whole network. When this process finishes we plan to submit a new paper wherein information regarding the uncertainties and the quality of the dataset will be also included.

We went through the manuscript very carefully and tried to improve the language. Furthermore, we tried to address each of the specific comments of the reviewer#2.

**Specific comments:**

**l. 34-38: Is it really so that people are adapted to proper sun-exposure behaviour? Do you have an example? I think there exist still many problems at areas of high exposure, e.g. cataracts in Tibet.**

**Reply**

According to the reference provided in the original version of the manuscript, the skin coloration of indigenous people at different latitudes is a result of the evolutionary process and depends on the levels of the available UV radiation. So, what was written in the original version was not accurate. The manuscript has been modified as follows (lines 36 - 42):

"Various living organisms, including humans, have been slowly adapted through centuries to the levels of UV radiation at the place where they live. For example, the skin coloration of indigenous people at different latitudes is a result of the evolutionary process and depends on the levels of the available UV radiation (Jablonski and Chaplin, 2000). However, sun-exposure behaviours of humans are still not optimal in many cases, being responsible for health issues directly related with over- or under-exposure to solar UV radiation. Malignant melanoma (Moan et al., 2008) and cataracts (Taylor et al., 1988;Bourne et al., 2013) are common problems caused by the excessive exposure to solar UV radiation, while hypovitaminosis D (Juzeniene et al., 2011) is a common problem caused by the inadequate exposure to UV radiation."

**l. 46: Please check if Solomon et al. 1986 describes the Northern ozone depletion. If not, please find a reference for the Arctic ozone depletion and check the decade of the first signs of arctic spring-time ozone depletion.**

**Reply**

As the reviewer correctly noticed, Solomon et al. 1986 does not describe the Arctic ozone depletion. Thus, appropriate references were added. As now also referred in the manuscript, the first signs of Arctic spring-time ozone depletion were found at the beginning of the 1990s (see lines 55 - 56).

**l. 69: "for SZAs below 75◦ and wavelengths below 305 nm." Do you mean ABOVE 305 nm?**

**Reply**

Corrected (line 84)

**l. 100: "extremely high levels of the UV irradiance" , please add what are the extreme values in UV index (some number).**

**Reply**

The phrase "of 11 or more" has been added to clarify which values of the UV index we mean. Furthermore, we cite a study (Vanicek et al., 2000) wherein the reader can find more information for the UV index. New information is in line 117.

**l. 132: within less THAN 0.5 ◦C?**

**Reply**

"than" has been added to the sentence.

**l. 134: please discuss whether the humidity difference (10% in winter, 60% in summer) can introduce errors/uncertainties in measurements.**

**Reply**

As explained now in the manuscript, the variations of relative humidity do not have any significant effect on the response of the instrument (see lines 153 - 154).

**l. 164: at Saint Christophe -> at AAO?**

**Reply**

Done.

**l. 171: Do you have a web page for UV index prediction? If yes, please show the web address.**

**Reply**

A link to the web-page has been added (line 195)

**l. 186: If the Instrument characterization and measurement corrections of Section 3 are only about the reference spectroradiometer, please indicate it in the header of the Section 3**

**Reply**

The header has been changed to: "3 Characterization of the Bentham5541 and correction of the measurements"

**l. 194: Please add a sentence or two explaining what is dark current and from where does it come from**

**Reply**

A sentence explaining what dark current is has been added (lines 221 - 223).

**l. 208: Teflon diffuser: In line 146 you write that the instrument has a quartz diffuser?**

**Reply**

What was written in line 146 was a typo. The instrument has a Teflon diffuser. Now it has been corrected.

**l. 224: Eq.3. cf=**

**Reply**

Equation 3 has been changed, as both reviewers suggested. The letter "c" is used instead of "cf".

**l. 236-238: It is unclear what you actually did and how did you end up to the result. Please rephrase.**

**Reply**

We rephrased the document making clearer what was done in order to end up with this result (lines 271 - 277).

**Section 3.2.: Please explain what is the typical effect (in %) of the temperature correction of the Teflon diffuser for a dataset during a winter month and a summer month.**

**Reply**

It is up to 3% in winter, and up to 1% in summer ( information added in lines 266 - 268)

**l. 278: For which wavelengths are the results showed? Any wavelength dependency in the angular response?**

**Reply**

The wavelength dependency of the angular response was already discussed at the last paragraph of section 3.3.1 already in the previous version of the manuscript. However it has been updated and moved earlier at the same paragraph. The results shown in this paragraph are for 320 nm. This information has been also added to the manuscript.

**l. 300: Header: Modelling the errors due to what, angular response? Please change the header to be more explanatory.**

**Reply**

**The header has been changed to: "3.3.2 Modelling the errors due to angular response"**

**l 309: "Simulations for Davos were performed in order to show that at such high altitudes the error becomes more important", please explain why this is the case.**

**Reply**

An explanation has been added as suggested by the reviewer (lines 400 - 404).

**Section 3.5.1., l.370. I don't really understand how you can address this point by 200W lamp measurements as your 200W lamp has always the same intensity/irradiance, hasn't it? Shouldn't you vary the intensity of the lamp in order to address the point 2?**

**The same applies for Section 3.5.2., or do you mean that your PMT has a "memory"?**

**Reply**

Section 3.5 was obviously confusing. It has been re-written in a clearer way after the suggestions of both reviewers. It is now explained in a clearer way, that when PMTs of the same type as that of Bentham5541 produce photocurrents much higher than their optimal operational range, their responsivity changes either temporally (indeed it is like the PMT has a "memory").

**Section 4.1.: How do you take into account the possible drift of your 200W lamps? Do you rotate them, as you have three lamps?**

**Reply**

The relative information has been added to the manuscript (lines 505 - 507).

**Section 4.2.4: It is a little bit diffucult to follow what has happened during each period. Anyway, the message is unclear: is the Level 2 time series homogeneous? If yes, which calibration scale has been used? Have you used a step-wise calibration change in calibration: change after each calibration? Have you taken into account that the instrument's reponse may have changed slowly between two calibration, and have this been taken into account e.g. by linear interpolation between two calibrations?**

**Reply**

Section 4.2.4 has been shortened and re-written in a clearer way. All information asked by the reviewer has been added in a short paragraph at the end of the section.

**Fig 8. Please check the Figure Caption.**

**Reply**

We found an error in the caption, which has now been corrected. The sentence:

**"Shaded area represents ±2% from perfect agreement between Bentham5541 and QASUME "**

Has been replaced by:

"Shaded grey areas represent the combined expanded uncertainty of the Bentham5541 Level 2 and the QASUME spectra for each intercomparison"

**l. 630, This is not true for all years (2006 and 2013).**

**Reply**

The phrase "for most years" has been added before the phrase "There is a clear improvement of the results when the Level 2 dataset is used." (line 784)

**Section 5.2.5. I miss a table with all the uncertainty sources contributing the overall uncertainty (like you have in Table 1 for radiometric uncertainty).**

**Reply**

Table 3 has been added to the document.

[revised manuscript text omitted]

---

## Author Response (AR2)

We would like to thank the editor for his efforts. We took into account his suggestions and applied the corresponding technical corrections to the manuscript. Analytical replies to all editor's comments are provided in the following.

**Comment**

**Lines 48, 49: Correct about ozone but counter to most medical advice about dangers of UV-B vs UV-A?**

Reply

At this sentence we state that the more energetic photons with shorter wavelengths in the UV-B range are more effective than the photons with longer wavelengths (in UV-B and UV-A) on causing acute and chronic diseases. We refer to single photons. The editor has certainly a point since the UV-A component is much larger than the UV-B component of the solar irradiance that reached the surface and its role is not negligible.

Nevertheless, we do not claim that photons in the UV-A region are not effective. What we write is in agreement with what is referred in the existing bibliography, i.e. that photons in the UV-A region are also effective, but not as effective as the photons in the UV-B region. Since they are much more than the photons in the UV-B the role of UV-A is in many cases more important than the role of UV-B irradiance.

We added the following sentences in the manuscript (lines 51 – 54) in order to make everything clearer:

"Photons in the UV-A spectral region (wavelengths 315 – 400 nm) are absorbed less effectively by ozone relative to the photons in the UV-B region. Thus, the overall UV-A irradiance reaching the earth surface is much larger (by more than two orders of magnitude) than the UV-B. "

**Comment**

**Because their setup measures 290 to 500 nm, these authors get most of UV-B (nominally 280-315), all of UV-A (nominally 315-400) plus - as they note - some violet/blue fraction of visible? Please check for accuracy and consistency of all uses of and wavelength definitions of UV-B and UV-A?**

Reply

We added the range of UV-B radiation by definition (280 – 315 nm) in line 48, as well as the corresponding UV-A range (315 – 400 nm) in line 52. We added one more sentence relative to the previous version of the manuscript where we explain that all photons with wavelengths shorter than 290 nm are absorbed by ozone in the atmosphere and never reach the earth surface.

We also added a new sentence at line 128 clarifying that we do not perform measurements below 290 nm (although it is possible) because there is no solar irradiance with such short wavelengths reaching the Earth's surface. Thus we measure all UV-B irradiance by performing measurements in the range 290 – 315 nm instead of 280 – 315 nm.

**Comment**

**Later the authors describe data in terms of CIE-based erythemal irradiance (in some cases accompanied by UV-A) and various plant or animal effective doses, which seem much more relevant but - see below - reader never finds a definition of the CIE acronym nor a description of relevant wavelengths or exposures.**

Reply

A definition of the acronym CIE has been added in line 541.

**Comment**

**Line 147: Use of acronym QASUME here well before definition of the acronym. Same problem occurs for AOD, introduced later in the document without any definition. Please check all acronyms for accuracy, proper definition at first usage, etc.**

Reply

The definition of QASUME has been moved from line 197 to line 133 (first time when the acronym QASUME is used)

The definition of AOD has been added to line 335 (first time when the acronym AOD is used).

[revised manuscript text omitted]